# High fidelity DNA ligation prevents single base insertions in the yeast genome

Jessica S. Williams[1,3], Scott. A. Lujan [1,3], Mercedes E. Arana[1,3], Adam B. Burkholder[2], Percy P. Tumbale[1], R. Scott Williams [1] & Thomas A. Kunkel [1] ✉

Finalization of eukaryotic nuclear DNA replication relies on DNA ligase 1 (LIG1) to seal DNA nicks generated during Okazaki Fragment Maturation (OFM). Using a mutational reporter in *Saccharomyces cerevisiae*, we previously showed that mutation of the high-fidelity magnesium binding site of LIG1[Cdc9] strongly increases the rate of single-base insertions. Here we show that this rate is increased across the nuclear genome, that it is synergistically increased by concomitant loss of DNA mismatch repair (MMR), and that the additions occur in highly specific sequence contexts. These discoveries are all consistent with incorporation of an extra base into the nascent lagging DNA strand that can be corrected by MMR following mutagenic ligation by the Cdc9-EEAA variant. There is a strong preference for insertion of either dGTP or dTTP into 3−5 base pair mononucleotide sequences with stringent flanking nucleotide requirements. The results reveal unique LIG1[Cdc9]-dependent mutational motifs where high fidelity DNA ligation of a subset of OFs is critical for preventing mutagenesis across the genome.

During DNA replication, the two new DNA strands are synthesized asymmetrically, with the leading strand synthesized in a continuous fashion and the lagging strand synthesized as a discontinuous series of DNA fragments known as Okazaki fragments (OFs). OFs are initiated by DNA polymerase alpha (Pol α)-primase and then further extended by DNA polymerase delta (Pol δ) in cooperation with the sliding clamp PCNA[1,2]. When Pol δ reaches the 5′-end of a downstream DNA fragment, it performs nick translation/strand displacement synthesis to displace the 5-RNA-DNA fragment and generate a 5′-flap that must be removed to generate a DNA nick containing ligatable 5′-phosphate and 3′-hydroxyl DNA ends[3–7]. When the length of the flap is short, it can be cleaved by the flap endonuclease, Fen1[RAD27]. Flaps that escape cleavage by Fen1 can be lengthened by the Pif1 helicase and then cleaved by DNA2 to create a short flap that can be cleaved by Fen1[4,8,9]. The final step of Okazaki Fragment Maturation (OFM) is the ligation of nicks by DNA ligase 1 (LIG1) to generate a continuous lagging strand. Failure to

properly cleave 5′-flaps prevents strand displacement synthesis, impairs DNA ligation, and can lead to genome instability in the form of DNA breaks and mutations[10]. While much has been uncovered regarding the steps of OFM (reviewed in ref. 4), the importance of the fidelity of this process as it relates to strand displacement synthesis by Pol δ, flap processing by Fen1, and ligation by LIG1 is still emerging.

The accuracy of ligation by LIG1 partly depends on a "high-fidelity (HiFi)" site located 3–4 bases upstream of the site of catalysis that involves a Mg²⁺-reinforced DNA binding mode scaffolded by two conserved glutamate residues[11]. When mutated to alanines (LIG1/Cdc9-EEAA), the enzyme performs mutagenic ligation[11,12]. In the budding yeast *URA3* reporter gene, the *cdc9-EEAA* mutant displays a unique spontaneous mutator phenotype dominated by single-base insertions in homonucleotide runs. This rate is further increased upon *RAD27* deletion and is also elevated in the absence of *MSH2*[12], which initiates DNA mismatch repair (MMR) of replication errors introduced by the

[1]Genome Integrity and Structural Biology Laboratory, National Institute of Environmental Health Sciences, US National Institutes of Health, Department of Health and Human Services, 111 TW Alexander Drive, Research Triangle Park, NC 27709, USA. [2]Office of Environmental Science Cyberinfrastructure, National Institute of Environmental Health Sciences, US National Institutes of Health, Department of Health and Human Services, 111 TW Alexander Drive, Research Triangle Park, NC 27709, USA. [3]These authors contributed equally: Jessica S. Williams, Scott. A. Lujan, Mercedes E. Arana. ✉e-mail: kunkel@niehs.nih.gov

replicative polymerases during both leading and lagging strand synthesis[13–15]. These results support a model in which single-base insertion mutations resulting from DNA polymerase slippage during strand displacement synthesis are the consequence of low-fidelity ligation in the *cdc9-EEAA* mutant that is synergistically increased upon loss of MMR and/or Fen1-dependent flap processing. Consistent with this model, the *cdc9-EEAA msh2Δ rad27Δ* triple mutant is inviable[12]. Genome stability requires efficient and accurate ligation of the abundant DNA nicks produced during OFM, which are generated at replication origins and about every 250 base pairs during lagging strand replication.

Here we investigate the genome-wide mutation specificity of the low-fidelity LIG1 mutant in the presence or absence of MMR. Our results reveal that the genome mutation spectrum is dominated by single base insertions in short homopolymer runs that arise during lagging strand synthesis. These occur in specific contexts with well-defined hotspots and strong strand bias. Taken together, the discovery of a mutable sequence motif that requires high-fidelity DNA ligation and MMR to prevent +1 mutations demonstrates the importance of DNA ligase fidelity as a critical determinant of lagging strand replication fidelity, and strongly suggests that high-fidelity OFM, as mediated by MMR and OFM flap processing, are critical for viability and preventing single base addition mutations.

## Results

### Mutagenic ligation by Cdc9-EEAA creates additions of G and C base pairs

Using a *URA3* mutational reporter integrated into the yeast genome adjacent to the *ARS306* origin of replication in orientation 1 (*URA3-OR1*), we previously showed that the *cdc9-EEAA* mutations in *S. cerevisiae* LIG1 result in addition of a single base pair in mononucleotide runs of C•G and A•T base pairs[12], and that these addition rates are synergistically increased when *MSH2* is also deleted. Although the *URA3* gene contains 17 runs of C•G base pairs, the rates of these addition mutations were primarily observed at three hotspots, positions 344, 564, and 612, all of which are runs of three consecutive C•G base pairs flanked by a T•A base pair (Supplementary Fig. 1 and ref. 12). By combining knowledge of the strandedness of the *URA3-OR1* reporter gene from DNA polymerase error signatures[2,16,17], along with ribonucleotide incorporation studies[18–21] that identify the leading and lagging DNA strands, this specificity suggested that addition mutations at these hotspots resulted from incorporation of an extra C into the lagging DNA strand. Biochemical and structural analyses of mutant LIG1-EEAA bound to a bulged DNA substrate corresponding to the hotspot at position 344 show how the enzyme accommodates the flipped nucleotide into a pocket generated by the mutation[12].

On this basis, we began the present study by measuring spontaneous mutagenesis in *cdc9-EEAA ± MSH2* strains when the *URA3* reporter gene was inserted in the opposite orientation, Orientation 2 (*URA3-OR2*). Overall mutation rates in *OR1* and *OR2* were similar ($4.2 \times 10^{-8}$ versus $2.3 \times 10^{-8}$) in an MMR-proficient background (Supplementary Table 2), and the *URA3-OR2* mutation spectrum was again dominated by single base additions at the same hotspots (Supplementary Fig. 1a). In contrast, many of the other GGG runs present were not changed. Similar overall mutation rates were also observed in the *cdc9-EEAA msh2Δ* strain when comparing *URA3-OR1* to *-OR2* ($300 \times 10^{-8}$ vs. $590 \times 10^{-8}$; Supplementary Tables 2, 3), and the rate of single base insertions, particularly at the hotspots, was even greater for the *URA3-OR2* reporter than had been observed for *-OR1* (Supplementary Fig. 1b). Because the sequence context of the lagging strand template is flipped in the two orientations, these results suggest that the extra nucleotide responsible for the *cdc9-EEAA*-dependent additions can be either a C or a G. Thus, in accordance with the model for accommodation of an extra nucleotide due to structural plasticity imparted by the cavity created by mutating the high-fidelity metal-binding site in

**Table 1 | Genomic mutation rates in the MMR-proficient and –deficient strains expressing wildtype or the Cdc9[LIG1]-EEAA mutant**

| Strain | Outgrowths | Generations | Mutations | µ (/Gbp/generation) | SD |
|---|---|---|---|---|---|
| Wildtype | 39 | 53,6760 | 255 | 0.21 | 0.077 |
| *msh2Δ* | 5 | 4260 | 4010 | 43 | 8.5 |
| cdc9-EEAA | 43 | 129,000 | 984 | 0.35 | 0.081 |
| cdc9-EEAA *msh2Δ* | 36 | 10,369 | 47,076 | 200 | 27 |

Genomic mutation rates from mutation accumulation and whole genome sequencing (WGS) of the indicated yeast strains. Rates are per Gbp per generation. µ genomic mutation rate; SD standard deviation. Source data are provided as a Source Data file.

Cdc9, the bulged nucleotide could theoretically be either a C or a G[12], but with a two to seven-fold preference for G when comparing the +1 rates at the hotspots in URA3-OR1 versus -OR2 for the *cdc9-EEAA msh2Δ* strain (Supplementary Table 3), and with a preference for G•C-runs flanked by a T•A base pair (Supplementary Fig. 1).

### High-fidelity DNA ligation avoids single base addition mutations across the genome

To expand the mutational target 15,000-fold, we next determined the *cdc9-EEAA* mutation profile in the presence or absence of MMR by whole genome sequence analysis. Overall and specific genomic mutation rates were determined as previously described in refs. 13,22,23, and the *cdc9-EEAA* and *cdc9-EEAA msh2Δ* mutants were then compared to published data for wildtype and *msh2Δ* control strains[13,23,24]. Spontaneous mutations were accumulated in multiple outgrowths on solid rich medium at 30 °C. Each outgrown descendent line underwent single-cell bottlenecks roughly every 30 generations to fix the accumulated mutations within the population. Samples from initial and final populations were sequenced and compared to discover which mutations arose during the experiment. A total of 984 newly accumulated mutations were identified for the *cdc9-EEAA* single mutant, and 47,076 new mutations were identified in the *cdc9-EEAA msh2Δ* double mutant (Table 1). The overall genomic mutation rate (µ) in the wildtype and *cdc9-EEAA* mutant strains differed slightly but significantly (1.6x; $0.21 \pm 0.012$ vs. $0.35 \pm 0.012$ Gbp$^{-1}$gen.$^{-1}$, respectively; $p = 5.0 \times 10^{-10}$). However, there was a 950-fold increase in the genomic mutation rate in the *cdc9-EEAA msh2Δ* double mutant strain when compared to the wild-type strain ($200 \pm 4.4$ vs. $0.21 \pm 0.012$ Gbp$^{-1}$gen.$^{-1}$) and a 4.7-fold increase compared to the *msh2Δ* single mutant ($200 \pm 4.4$ vs. $43 \pm 3.8$ Gbp$^{-1}$ gen.$^{-1}$; $p = 5.8 \times 10^{-6}$) (Table 1).

Mutation rates and rate ratios were then calculated for base pair substitutions (bps), single-base insertions and deletions (±1 bp indels) and >1 bp indels (Table 2 and Fig. 1). Single-base insertions and deletions were elevated in the *cdc9-EEAA* strain compared to wildtype (gray bars in Fig. 1a), and rates for all three mutation types were elevated in the *cdc9-EEAA msh2Δ* strain compared to the *msh2Δ* strain (orange bars). Rates increased little, if any, for base pair substitutions and multi-base indels in both MMR$^+$ and MMR$^-$ backgrounds. Rates for 1 bp indels, especially 1 bp insertions, were greater in the *cdc9-EEAA* mutant than in *CDC9$^+$* strains (Fig. 1a) in both the MMR$^+$ and MMR$^-$ backgrounds. These 1 bp insertions were widely distributed across the genome (Supplementary Fig. 2 and below). Single-base insertion rates were 20-fold higher for *cdc9-EEAA* compared to wildtype (*CDC9$^+$*), and 24-fold higher for *cdc9-EEAA msh2Δ* compared to *msh2Δ* (Table 2). A•T insertions were more numerous than G•C insertions (Supplementary Data 1). After correcting for the low GC content of the *S. cerevisiae* genome, G•C insertion rates exceeded A•T insertion rates in both MMR-proficient and -deficient *cdc9-EEAA* strains (Fig. 1b). The *cdc9-EEAA* mutant had a +G•C

rate 39-fold higher than wildtype, and the *cdc9-EEAA msh2Δ* strain had a +G•C rate 120-fold higher than the *msh2Δ* mutant. Interestingly, A•T insertion rates are also elevated for the *cdc9-EEAA* mutant in both the presence and absence of MMR (Fig. 1b). Taken together, the genomic mutation rates and specific mutation rate data provide evidence for the critical roles that high-fidelity DNA ligation and MMR play in preventing +1 mutations in mononucleotide runs of both G•C and A•T base pairs across the yeast genome.

### Cdc9-EEAA selectively ligates single base insertion intermediates in short mononucleotide runs

The yeast genome contains numerous homopolymer tracts of various lengths and in different flanking nucleotide sequence contexts. To examine the effect of sequence context on indel rates, we next plotted the genome-wide rate for the addition of G•C and A•T base pairs as a function of homopolymer run length. In the wild-type strain (Fig. 2a), single-base additions are rare, and are either not observed or, in rare instances (G•C in orange; A•T in red), are observed right at the limits of detection (dashed lines). In contrast, in the *msh2Δ* strain (Fig. 2b), the rates for the addition of G•C and A•T base pairs are significantly higher at most run lengths. This fact is consistent with the known critical role of MMR in efficiently correcting a wide variety of DNA replication errors across the genome[13,14], including single base insertions.

In both the wildtype and *msh2Δ* backgrounds, EEAA mutations in DNA ligase 1 further increased G•C and A•T insertion rates (Fig. 2c, d, respectively). Both +G•C rates and +A•T rates were well above the detection limit in homopolymers of <15 bp. +G•C rates exceeded +A•T rates for most homopolymer lengths. +G•C rates increased most in runs of three to five base pairs, while +A•T rates increased most in runs of four to five base pairs (Fig. 2e, f). In the yeast genome, short

homopolymers are mostly found within open reading frames and vastly outnumber longer homopolymers, with counts decreasing exponentially as length increases from one to eight base pairs[25]. Thus, regardless of MMR status, the vast majority of insertions caused by the *cdc9-EEAA* variant are in three to four bp runs of G•C or four to five bp runs of A•T. When the number and positions of the +G•C insertions in three base pair homopolymer runs were plotted across the 16 yeast chromosomes (Fig. 2g), a wide distribution was observed, with some definite hotspots (pink in color). Taken together, these data reveal that G•C and A•T insertion rates are selectively increased in short homopolymers during mutagenic ligation by Cdc9-EEAA. They also clearly indicate that ligation by wild-type DNA ligase 1 is much more accurate.

### +G mutagenesis is biased to the lagging strand across the genome

Next, we investigated whether the DNA ligase-mediated insertion events were arising during the synthesis of the leading or lagging DNA strand. Single-base insertion events occur when an extra base is incorporated into the newly synthesized strand during replication. For example, if G insertions are preferentially made during lagging strand synthesis (shown in green in Fig. 3a, b), then they will be converted into mutations in the subsequent round of replication. For the genomic sequencing data, mutations are reported in the top strand, and a bias for incorporation of an extra G would cause C insertions to be preferentially called to the right of origins and G insertions to be preferentially called to the left. Using the sequencing results from the *cdc9-EEAA msh2Δ* strain, the fraction of G (orange) and C (blue) insertions were mapped between all adjacent origins. The resulting X-pattern indicates a preference for G insertion in the nascent lagging strand (Fig. 3c). The data can also be analyzed to determine the fraction of the bottom strand synthesized by Pols α and δ during lagging strand synthesis using ribonucleotide mapping data[26]. As this fraction approaches 1, the top strand G insertion rate decreases and the top strand C insertion rate increases (Fig. 3d). The fraction of G•C insertions also follows this pattern, as it decreases for +G and increases for +C (Fig. 3e). The y-intercept thus suggests a 7.4-fold preference for G insertions over C insertions during lagging strand synthesis in the *cdc9-EEAA msh2Δ* strain.

### High-fidelity DNA ligation and MMR prevent insertion mutagenesis in specific sequences

The preference for G insertions over C insertions for the *cdc9-EEAA msh2Δ* strain (Fig. 3d) is decreased in the presence of MMR (Fig. 4a). This suggests that MMR may have a bias for preferentially repairing bulged G bases in the nascent lagging strand. As the homopolymer

**Table 2 | Genomic mutation specificity in the MMR-proficient and –deficient strains expressing wildtype or the Cdc9^LIG1-EEAA mutant**

| Rate (/Gbp/ generation) | Base pair substitutions | 1 bp deletions | 1 bp insertions | >1 bp deletions | >1 bp insertions |
|---|---|---|---|---|---|
| Wildtype | 0.18 | 0.007 | 0.007 | 0.0067 | 0.019 |
| *msh2Δ* | 5.8 | 26 | 2.9 | 8.5 | 1.6 |
| *cdc9-EEAA* | 0.16 | 0.024 | 0.14 | 0.0079 | 0.019 |
| *cdc9-EEAA msh2Δ* | 9.7 | 98 | 71 | 16 | 5.4 |

Mutation rates for specific mutation classes are displayed. See the Methods section for a description of rate calculations.

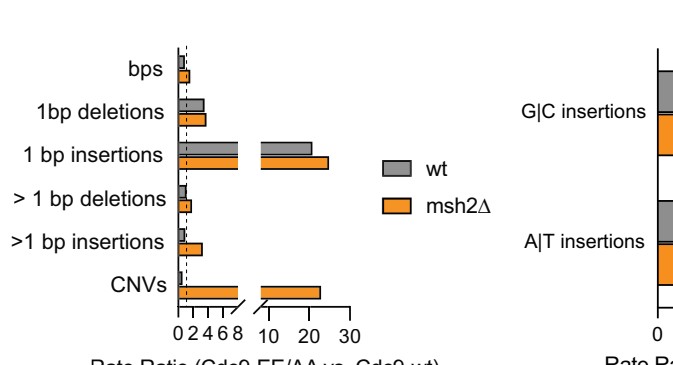

**Fig. 1 | High-fidelity DNA ligation is critical for avoiding +1 mutations across the genome. a** The mutation rate ratios of the various mutation classes were calculated as the rate in a strain expressing Cdc9-EEAA divided by the rate in a strain expressing wild-type *CDC9* (+/− MMR). The dotted vertical line indicates a ratio of 1. **b** The specificity of 1 bp insertions was determined for +G•C or +A•T insertions. The rate ratios were calculated as in (**a**). Source data are provided as a Source Data file.

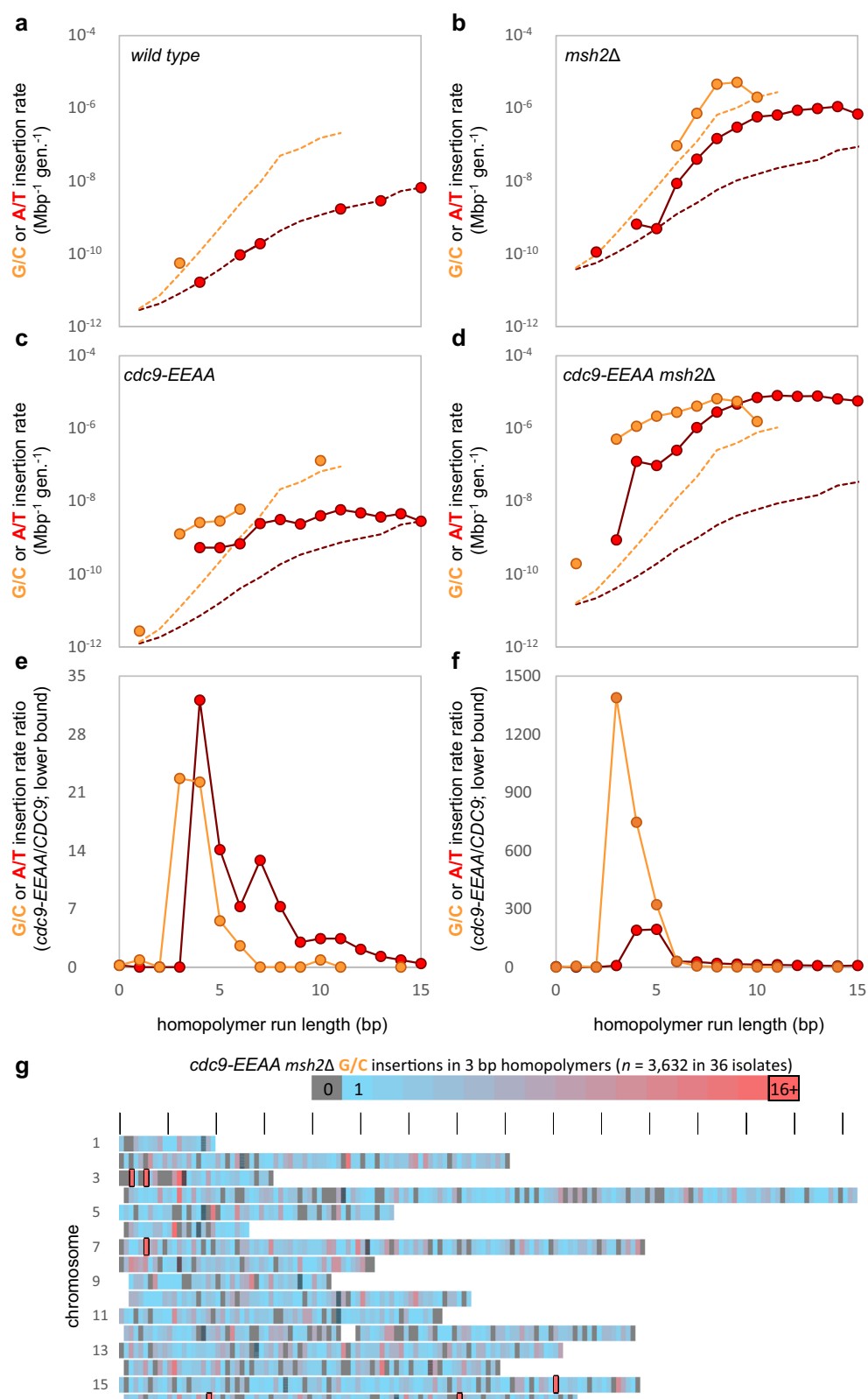

**g** *cdc9-EEAA msh2Δ* **G/C** insertions in 3 bp homopolymers (*n* = 3,632 in 36 isolates)

length increases from three to five base pairs in the MMR-deficient *cdc9-EEAA* mutant, the preference for G insertions and the correlations between strandedness and insertion bias both decrease (Fig. 4b–d). The bias goes from 19-fold to 5.6-fold, and the $R^2$ from 0.97 to 0.75 (Supplementary Data 1 and Supplementary Table 4). Moreover, in the *cdc9-EEAA* mutant, 98% of inferred lagging strand +G mutations in three base pair homopolymer runs are preceded, in the direction of

synthesis, by template A (Fig. 4e and Supplementary Table 5). 89% are then flanked at the −2 position by a template pyrimidine (C or T). The same flanking preferences were present in four base pair template C-runs (80 and 68%, respectively; Supplementary Fig. 3) and in the *cdc9-EEAA msh2Δ* strain (Fig. 4f). The strength of the preference decreased with increasing homopolymer length (Fig. 4g, h). The inferred motifs from these analyses are illustrated in Fig. 4i.

**Fig. 2 | Low-fidelity ligation selectively increases insertion rates in short homopolymers. a** Insertion rates in the wild-type (wt) strain (*CDC9*+). Insertion rates (circles) in G•C (orange) and A•T (red) homopolymers are shown with their respective detection limits (dashed lines). Detection limits are the rates that would be implied by one observed insertion with a given base content and homopolymer length. The minimum detectable rate for a given context is the rate that would be calculated if one mutation were to be observed in one instance of that content. **b** As per (**a**), but for *msh2Δ*. **c** As per (**a**), but for *cdc9-EEAA*. **d** As per (**a**), but for *cdc9-EEAA msh2Δ*. **e** The sequence context-dependence of mutations caused by expression of the *cdc9-EEAA* variant in the presence of MMR. The ratio reported depends on the number of insertions observed in the *cdc9-EEAA* (N) and wt (*CDC9*+) (M) strains. If N > 0 and M > 0, the ratio uses both rates (*cdc9-EEAA/CDC9*+). If M = 0, the ratio is a lower bound estimate using the *CDC9*+ detection limit as the *CDC9*+ rate. If N = 0, the ratio is reported as 0. If there are no homopolymers of a given type and length, no ratio is reported. **f** As per (**e**), but for *cdc9-EEAA msh2Δ*. **g** Heatmap of G•C insertions in 3 bp homopolymers in the *cdc9-EEAA msh2Δ* strain across the 16 *S. cerevisiae* chromosomes (10 kbp bins). Unmapped bins are colored white. All others are colored by the number of observed mutations. Bins with no mutations are gray, bins with one mutation are blue, bins approaching the significance threshold are red, and intermediate counts transition between blue and red. Bins with counts exceeding the significance threshold have a black border (Šidák correction; each bin counts as a hypothesis tested; family-wise error rate = 0.05). Bins containing centromeres are crosshatched black. Source data are provided as a Source Data file.

A parallel analysis was performed for A•T insertions in the MMR-proficient and -deficient *cdc9-EEAA* strain. In the presence of MMR (Fig. 5a), the preference for T insertions (red) over A insertions (green) during lagging strand synthesis is greater than the preference for G relative to C insertions (3.9x for T versus 2.2x for G, see Fig. 4a). In the absence of MMR (Fig. 5b–d), the preference for T insertions and correlations between strandedness and insertion bias both decrease (bias from 8.3x to 1.3x and R2 from 0.948 to 0.045) as the homopolymer length increases from 4 to 6 bp (Fig. 5b–d). Although no sequence motif logo was identified for the *cdc9-EEAA* single mutant strain (Fig. 5e), there was a notable motif for lagging strand T insertions in four base pair runs in the *cdc9-EEAA msh2Δ* double mutant strain (Fig. 5f). Here, 63% of inferred lagging strand T insertions in four base pair runs occurred opposite template A-runs that are flanked by a 5′ template C (Fig. 5f and Supplementary Table 5). This motif disappeared with increasing template A-run length (Fig. 5g, h). Figure 5i shows a model of the inferred motifs in the context of replication.

Combining the *cdc9-EEAA msh2Δ* motifs leads to a model in which single base lagging strand insertions occur during OFM in contexts with a template super-motif of $5'-C_nA_m-3'$. The highest insertion rates were found in contexts where (n,m) = (3 to 5,1) or (1,4 to 5). This encompasses both the template RCCCAY (+G) and CAAAAY (+T) hotspots. These data suggest that high-fidelity ligation by DNA ligase 1, together with MMR, prevent many single base insertion errors that have been introduced by Pol δ during strand displacement synthesis within this motif. There are 2870 one-bp insertions in just under 200,000 three base pair G•C-runs. Given these 2870 insertions, it was expected that approximately 21 sites would be hit more than once due to random chance, but 429 insertion sites were observed more than once (Supplementary Data 1 and Supplementary Fig. 4a). Furthermore, 819 single base insertion events were observed in just under 3000 CRCCCAY (template) tracts. It was expected that 95 sites would be hit more than once due to random chance, but 169 multiple-insertion sites were observed (Supplementary Data 1 and Supplementary Fig. 4b). The hottest hotspots are exclusively observed in the *cdc9-EEAA msh2Δ* double mutant strain (not observed in wildtype or the single mutants) and significantly more hotspots were observed than would be predicted due to random chance. Many of the +G•C insertion hotspots across the 16 yeast chromosomes were due to individual sites that were each observed in many isolates (Fig. 2g), including a +G insertion on chromosome 15 that was seen in 13 independent isolates and is a perfect match for the CCCAY template motif.

The frequency of hotspots around genomic features was next analyzed to determine possible strong patterns using the 15 hottest hotspots. They are distributed throughout genes (n = 9; 60%) and intergenic sequences (n = 6; 40%). This means that they were found more often than expected in intergenic sequences, given that only around 15% of the *cerevisiae* genome is intergenic and G-run densities are slightly lower in intergenic sequences (Supplementary Fig. 5). When intergenic, the hottest hotspots are found between converging, diverging and parallel transcribed genes. The mean distance to the nearest origin is approximately 9 kb, similar to the distance to the nearest origin for randomly selected positions (~10 kb). For the hot-spots located within genes, there appears to be no relationship between the direction of transcription and the direction to the nearest origin, or to the timing of replication (Supplementary Fig. 5). Although the hotspot loci replication times are weighted slightly later, so too is the overall genome[13]. In summary, there appears to be no obvious pattern regarding hotspots and the level or direction of transcription, replication origin proximity, or replication time.

## Discussion

The data presented here lead to several interesting interpretations. The strong increases in single base addition rates observed across the genome in the *cdc9-EEAA* mutant are primarily observed in mono-nucleotide runs of three to five base pairs, with little or no increases observed in longer mononucleotide runs (Fig. 2e). This fact is consistent with the crystal structures of DNA ligase 1[11,12], which reveal that changing the two glutamate residues to alanines eliminates binding of a 'high-fidelity' magnesium ion located three to four base pairs upstream of the ligase active site. This is exactly the location where an extra base in the newly synthesized strand containing a three to five base mono-nucleotide run would bulge upon DNA strand slippage generated during OFM. This strongly suggests that loss of magnesium ion coordination in the Cdc9-EEAA mutant ligase allows ligation at the active site located at three to five base pairs downstream. Consistent with this interpretation, the EEAA mutant ligase has been demonstrated biochemically to remain functional but has lower ligation fidelity[11,12].

The above interpretation, in turn, leads to a second and related possibility, that the position of the single base additions primarily reveals a subset of sequence contexts across the genome where OF ligation occurs. DNA ligase 1 seals nicks to complete several different DNA transactions, including recombination and multiple types of DNA repair. However, the number of ligation events required for the completion of these reactions pales in comparison to the much larger number of ligation events required to complete nuclear DNA replication, which is estimated to occur once for every ~250 base pairs on the lagging strand of eukaryotic nuclear genomes.

OFs are reported to be sized according to the nucleosome repeat with ligation sites preferentially occurring near nucleosome dyads. This suggests a strong relationship between OF ligation and chromatin structure[27]. While we observed a significant number of +1 mutations in our *cdc9-EEAA+/−* MMR strains, these likely occur at a small subset of the total ligation events required to complete lagging strand replication. These single base insertions occur during lagging strand replication (Fig. 3) and are repaired by DNA MMR (Table 1), strongly implicating aberrant ligation of OFs by Cdc9-EEAA as the culprit for the addition mutations. This point is particularly relevant because among the many different types of single-base mutations that can be corrected by MMR, the mismatches that result in single-base additions in the *cdc9-EEAA* mutant occur at the highest rates (Table 1 and Fig. 3). Compared to wild-type cells, the efficiency of MMR repairing of these mismatches is 10,000-fold (Table 2), which is among the highest efficiency for any type of mismatch. Compared to the bulk of replicative

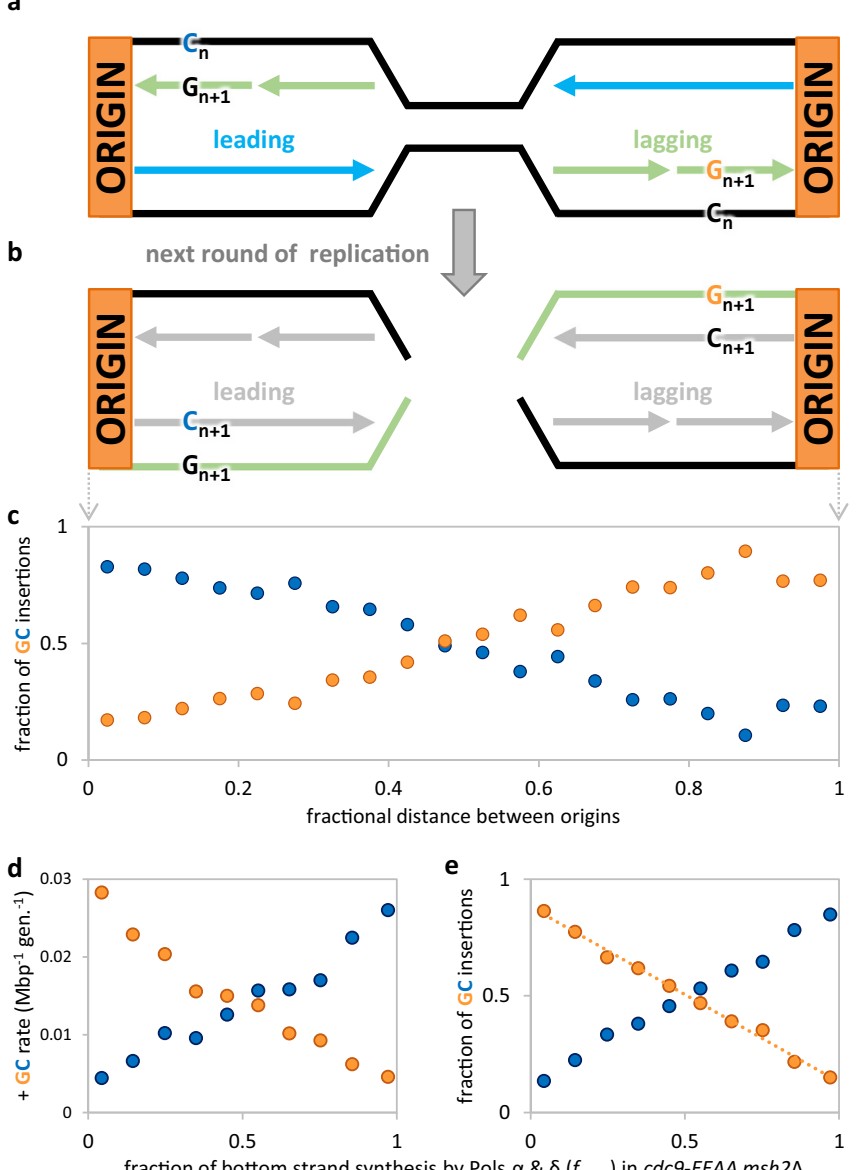

**Fig. 3 | Lagging strand G insertions are favored over C insertions in the low-fidelity DNA ligase mutant lacking MMR.** All measures are from the *cdc9-EEAA msh2Δ* strain. **a** A diagram of G insertions preferentially made during synthesis of nascent lagging strands. Thick lines denote parental (black) and nascent strands (arrows; leading in blue; lagging in green). **b** Insertion loops are converted into mutations when new DNA strands (gray) are synthesized in the subsequent round of replication. After sequencing, mutations are reported in the top strand reference frame. A bias for lagging strand G insertion loops would thus cause C insertions to be preferentially called to the right of origins and G insertions to be preferentially called to the left. The opposite would be true if C insertions were preferred in the nascent lagging strand. **c** When the fraction of G (orange) and C (blue) insertions is mapped between all adjacent origins, the resulting X-pattern indicates a preference for G insertion in the nascent lagging strand. **d** A better proxy for bottom lagging strand synthesis is the fraction of the bottom strand synthesized by Pols α & δ ($f_{BSαδ}$; a from ribonucleotide mapping[23]). As $f_{BSαδ}$ approaches 1, the top strand G insertion rate decreases, and the top strand C insertion rate increases. **e** The top strand G insertion fraction ($f_{TS+G}$) decreases linearly as $f_{BSαδ}$ increases ($f_{TS+G} = -0.751 \times f_{BSαδ} + 0.881$; $R^2 = 0.994$). The y-intercept suggests a 7.4-fold preference for G insertions over C insertions during lagging strand synthesis in the *cdc9-EEAA msh2Δ* strain. Source data are provided as a Source Data file.

DNA synthesis on "open" primer-templates lacking downstream DNA, the single-base additions observed in the *cdc9-EEAA* mutant are generated during OFM, which requires strand displacement DNA synthesis. This suggests that strand displacement synthesis may be more prone to strand slippage, an idea that can be explored further in future studies.

These results demonstrate the importance of accurate DNA synthesis by Pol δ during strand displacement synthesis and MMR to preserve genome integrity at the site of ligation during OFM. Kadyrova et al. (2015) provided evidence that 1 nucleotide flaps can be removed by DNA MMR in a *rad27Δ* strain, and failure to do so results in 1 bp

insertions[28]. They concluded that MMR and Rad27 act in overlapping pathways to protect from insertion mutations that arise during slippage of the template DNA strand. However, the importance of MMR to replication fidelity when flap processing by Fen1 is intact has been unclear. Our results demonstrate that both MMR and high-fidelity DNA ligation are critical for preventing +1 mutations during OFM, most likely due to slippage of the Pol δ-synthesized strand. This +1 error signature is distinct from what has been observed following whole genome sequencing of MMR-deficient yeast strains expressing wild-type Cdc9 (this study[13]). In an *msh2Δ* strain, many of the mutations are ±1 bp frameshifts that occur in long A•T homopolymers that are largely

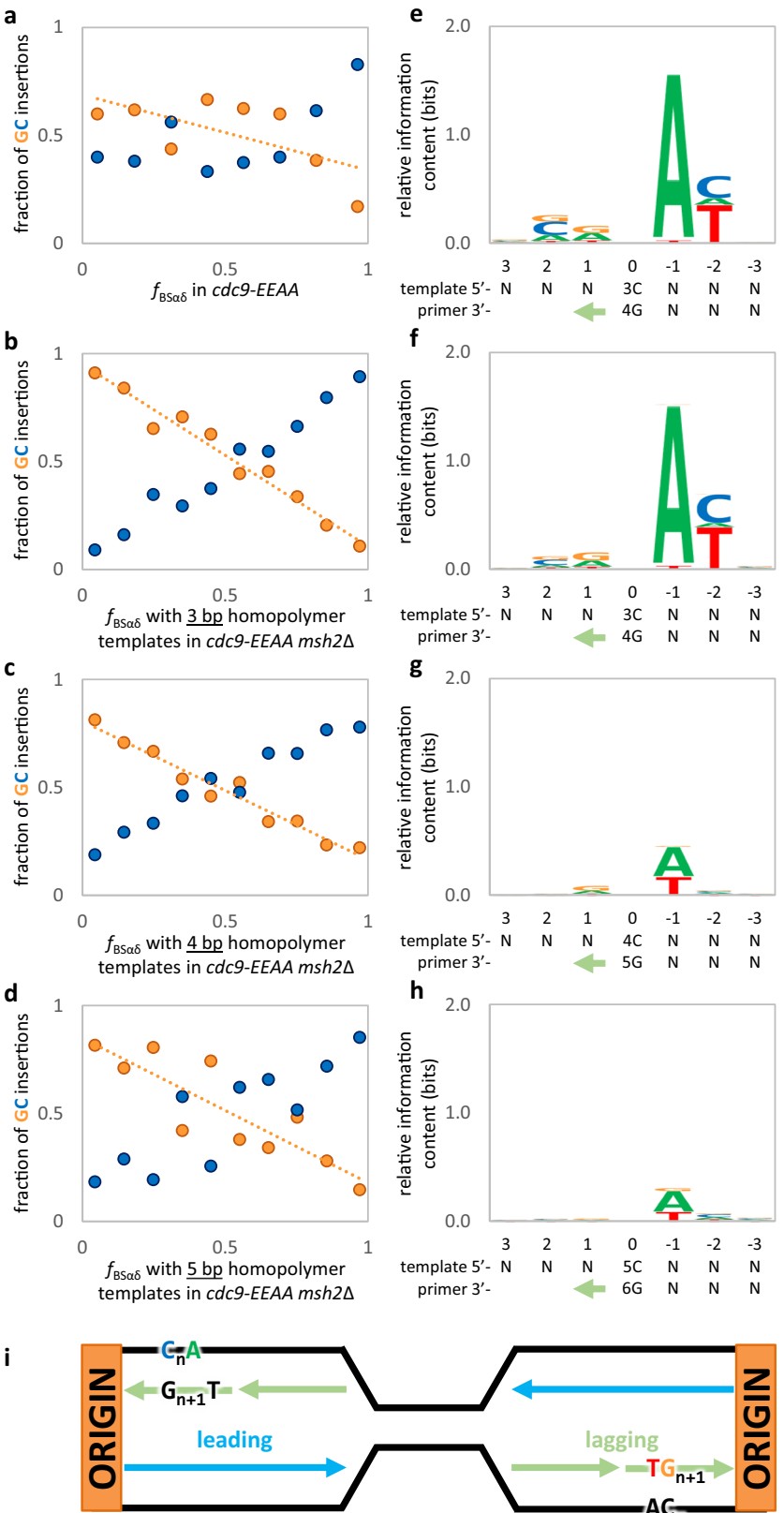

intergenic and not found within gene bodies. This is in stark contrast to the +G•C mutations observed in this study which are almost exclusively localized to gene bodies (Supplementary Fig. 5). The observation that small (≤4 bp) insertions have been shown to be more deleterious than small deletions in human genomes[29] suggests that mutations resulting from low-fidelity DNA ligation may be an important factor in evolution.

Furthermore, the reason for repeat lengths being different for T•A insertions than for C•G insertions (Figs. 4, 5) is intriguing and warrants further investigation. The data indicate an order of G > T > C > A, and both repeat tract length and the identity of the inserted nucleotide could be dictated by the fidelity of the DNA polymerase during synthesis, the fidelity of DNA ligase during nick sealing, or some

**Fig. 4 | In the low-fidelity *cdc9-EEAA* mutant, lagging strand G insertions have a template motif of 5′-C$_n$A-3′ that disappears, alongside bias, as homopolymer length increases. a** The preference for G insertions (orange) over C insertions (blue) during lagging strand synthesis is decreased in the presence of MMR (2.2x for *cdc9-EEAA* vs. 7.4x in *cdc9-EEAA msh2Δ*; see Fig. 3e). This suggests that MMR must have an opposite bias, preferentially repairing looped out G bases. **b–d** In the *cdc9-EEAA msh2Δ* strain, as homopolymer length increases from 3 to 5 bp, both the preference for G insertions and the correlations between strandedness and insertion bias both decrease (bias from 19x to 5.6x and $R^2$ from 0.969 to 0.748; Supplementary Table 4). **e–h** Sequence logos indicate preferred motifs for G insertions. **e** A sequence logo for all G insertions in the *cdc9-EEAA* stain. Using origin proximity to estimate strandedness, 98% of inferred lagging strand G insertions in 3 bp C-runs (*n* = 64) are found in runs that are followed by a template A, and 89% are then followed by template pyrimidines (C or T). These values drop to 80% and 68% in 4 bp template C-runs (*n* = 25; Supplementary Fig. 3). **f** Drawing from only the first and last 25% of each inter-origin space (see Fig. 4c), a nearly identical pattern was found in the *cdc9-EEAA msh2Δ* strain (98% A followed by 93% pyrimidine; *n* = 1488). **g, h** This motif disappeared with increasing C-run length (*n* = 697 and 347, respectively). **i** An illustration of the inferred motifs in the context of replication. Green arrows denote the direction of primer strand synthesis. Source data are provided as a Source Data file.

combination of both. These outstanding questions are of great interest and will require detailed biochemical analysis.

Mutation signature analyses have been informative in consideration of the mechanistic basis for DNA sequence changes in human tumors[30,31]. The data presented here offer the possibility that the unique LIG1-dependent mutational signature observed in the *cdc9-EEAA* mutant could be relevant to tumorigenesis. The two glutamate residues in LIG1$^{Cdc9}$ are strictly conserved throughout evolution and are not found in DNA ligases 3 and 4[11], ligases that are primarily involved in more specialized DNA ligation reactions that include DNA repair and somatic recombination (reviewed in ref. [32]). Cancer mutational signatures identified by Alexandrov et al. (2020) include 23 indel (ID) signatures involving the gain or loss of small fragments of DNA[31]. Of those extracted signatures, two are of unknown etiology (ID11 and ID16) that include only single base insertions of +G•C and +A•T, many of which are in short homopolymeric runs. In addition to mutations in LIG1, a single base insertion signature could also result from defects in other aspects of accurate OFM, including Pol δ-dependent strand displacement synthesis, PCNA, and/or Flap processing. For example, Njeri et al. (2023) demonstrated that acetylation enhances the ability of Pol δ to displace a downstream DNA fragment, and so it is possible that fidelity is also impacted. In addition to post-translational modifications, interaction with PCNA may also impact the fidelity of strand displacement synthesis by Pol δ.

The identification of +1 mutation hotspots and their template sequence motifs (5′-C$_n$A$_m$-3′) are striking. The extra base that is eventually added to the DNA (Fig. 6) may be initiated during strand displacement synthesis by a lagging strand replicase. One possibility is that this base is incorporated into the newly synthesized DNA strand because Pol δ has a propensity for slippage during strand displacement synthesis at T-G pyrimidine-purine junctions. This could result from less base stacking that eventually promotes the flipping out of an extra base into the pocket created in the Cdc9-EEAA mutant[12]. Another possibility is that an extra base may be incorporated by Pol α that is eventually converted into an addition by the well-documented flap equilibrium that can occur during OFM. The mutation hotspots may be related to such variables as DNA sequence context, secondary structure, and/or histone modifications. These are all possibilities that will be important to test in future studies to investigate the importance of high-fidelity OFM to genome stability.

## Methods

### Yeast strains
*Saccharomyces cerevisiae* strains are isogenic derivatives of strain Δ| (-2)|-7B-YUNI300 (*MATa CAN1 his7-2 leu2::kanMX ura3Δ trp1-289 ade2-1 lys2ΔGG2899-2900 agp1::URA3-OR1 or OR2*)[33], and relevant genotypes are listed in Supplementary Table 1.

### Spontaneous mutation rate and sequencing analysis
Mutation rate analysis was performed in strains containing the *URA3* reporter gene located adjacent to an efficient, early-firing replication origin, *ARS306*. Mutation rates and 95% confidence intervals (CI) were determined by measuring fluctuation analysis as described[34]. The *ura3* gene from single, independent 5-FOA-resistant (5-FOA$^R$) colonies was PCR-amplified and sequenced. Specific mutation rates were calculated by multiplying the fraction of that mutation type by the total mutation rate for each strain.

### Mutation accumulation for WGS
Yeast strains were subjected to single-cell bottleneck passages on YPDA agar plates (1% yeast extract, 2 % bacto-peptone, 250 mg/L adenine, 2% dextrose, 2% agar) as in ref. [13]. Each passage involved 2–3 days of growth at 30 °C to a colony diameter of between 2 and 3 mm, which was estimated to equal approximately 30 cell divisions. Samples were collected at time 0 and various timepoints to allow for the appropriate number of mutations to accumulate. Auxotrophic markers were checked throughout the passages, and samples were stocked at −80 °C before DNA isolation.

### Genome sequencing
Yeast genomic DNA was extracted and fragmented as in ref. [35]. Briefly, DNA was isolated using the Lucigen MasterPure Yeast DNA Purification kit (MPY80200). Sequencing libraries were prepared as in ref. [13] using the Illumina TruSeq DNA protocol. The libraries were quantified and then pooled for sequencing on a HiSeq 4000 platform (*cdc9-EEAA*) or an Illumina NovaSeq 6000 platform (*cdc9-EEAA msh2Δ*) (Illumina Inc.; 2 × 150 bp paired end reads).

### Sequencing data analysis and genomic mutation rates
DNA sequencing reads were mapped to the published L03 master reference sequence version 2[13,36]. Strain genotypes were confirmed from the mapping. Muver 1.2.4 was used to make variant calls as in ref. [24] with default settings, with the timepoint 0 being the ancestor and the final outgrowth samples as the descendants. Ploidy and lack of contamination were confirmed by allelic fraction plots (see below). Initial and final loci with significantly different allele distributions indicate mutations that arose during intervening cellular generations. Mutation rates were calculated as previously reported[26]. Briefly, for a given mutation type and context, the mutation rate is equal to the mutation count divided by the number of elapsed cellular generations and the number of base pairs where such a mutation could occur.

Mutation calls were compared between isolates to protect against sample switches/duplications, and incorrect ancestor (timepoint 0) assignment. One *cdc9-EEAA* lineage was found to have the wrong ancestor assignment and was excluded from the analysis. Two forked lineages were discovered (*cdc9-EEAA msh2Δ*, TAK2697/2699, and TAK2721/2722). To avoid double-counting, shared mutations were filtered from one member of each pair (TAK2697 and TAK2721), and their generation counts were set to the approximate point of divergence (127 and 192 generations before the end of the experiment, respectively).

Diploid strain construction using multiple independent approaches was unsuccessful, so haploid strains were constructed for passaging and mutation accumulation. However, all *cdc9-EEAA msh2Δ* lines were found to have acquired diploidy either before or during the

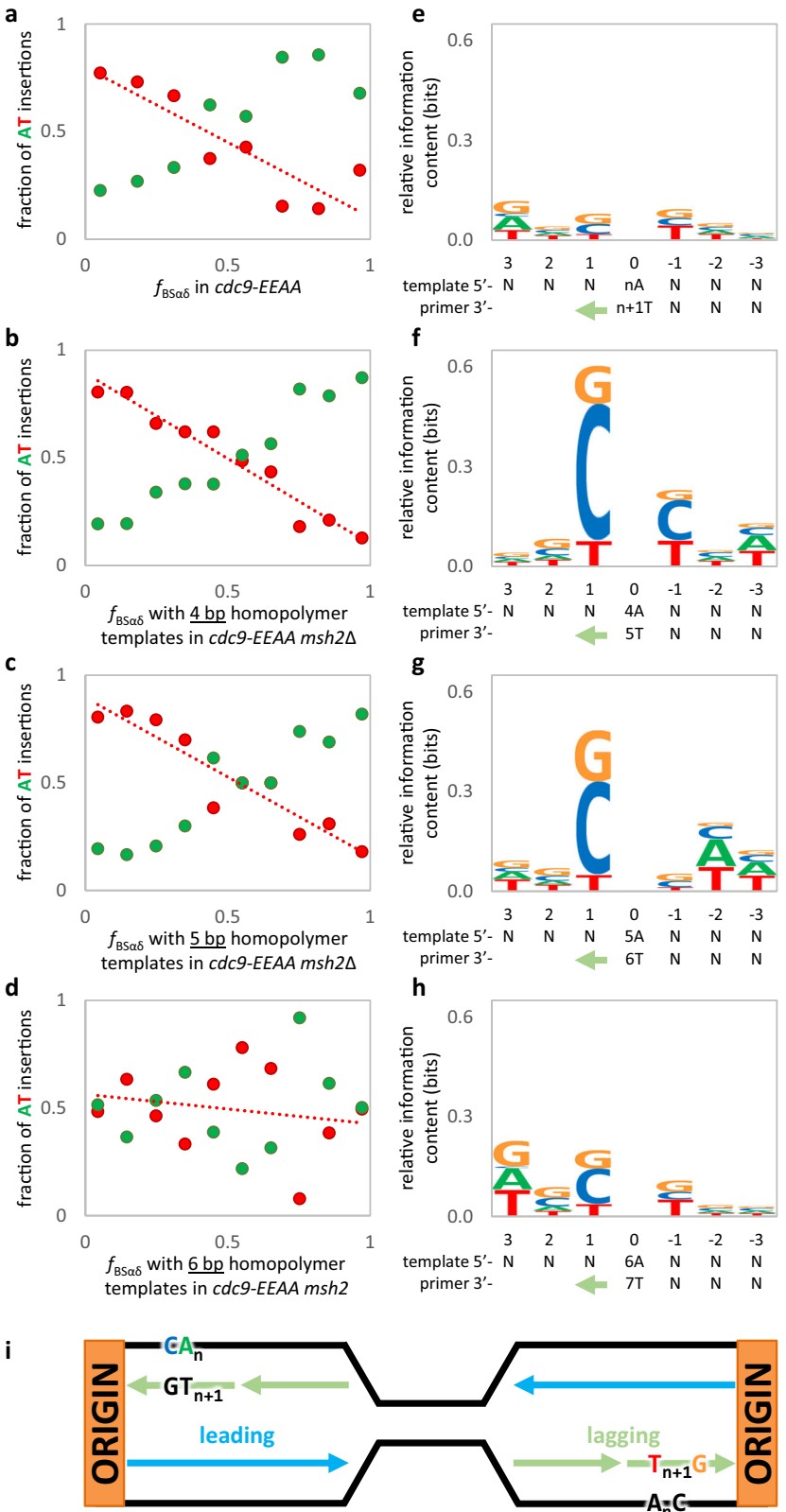

mutation accumulation experiment. Muver diploid parameters were thus used for all calls. Where multiple alleles appear to have changed in the same way at the same locus, muver reports each allele separately but flags all but one as due to potential loss of heterozygosity (LOH) (i.e., repeated point mutations, or LOH following a point mutation, or due to an initial ploidy lower than the one used for mutation calling; see Supplementary Data 1 for flag format). This default output is

reported here (Supplementary Data 1) and is used for hotspot detection. However, downstream calculations are simplified by assuming constant ploidy. Thus, to avoid underestimating diploid mutation rates in all other calculations, flags were reverted for these loci (Supplementary Data 1, listed as "cdc9-EEAA msh2Δ dubct").

For a lineage in which ploidy increases from haploid to diploid ($i$ = hap. and $i$ = dip., respectively), the mutation rate ($\mu_i$) is a function

**Fig. 5 | In the low-fidelity *cdc9-EEAA* mutant, lagging strand T insertions have a template motif of 5′-CAₘ-3′ that disappears, alongside bias, as homopolymer length increases. a** In the presence of MMR, the preference for T insertions (red) over A insertions (green) during lagging strand synthesis is greater than the preference for G relative to C insertions (3.9x for T vs. 2.2x for G; see Fig. 4a). **b**−**d** In the *cdc9-EEAA msh2Δ* strain, as homopolymer length increases from 4 to 6 bp, both the preference for T insertions and the correlations between strandedness and insertion bias both decrease (bias from 8.3x to 1.3x and $R^2$ from 0.948 to 0.045). **e**−**h** Sequence logos indicate preferred motifs for T insertions. Green arrows denote

the direction of primer strand synthesis. **e** A sequence logo illustrates the lack of a motif for T insertions in the *cdc9-EEAA* strain. **f** There is a notable motif for lagging strand T insertions across from 4 bp template A-runs in the *cdc9-EEAA msh2Δ* strain. Drawing from only the first and last 25% of each inter-origin space (see Fig. 5c), 63% of inferred lagging strand T insertions in 4 bp A-runs ($n = 64$) are found in runs that are immediately preceded by a template C ($n = 599$). **g, h** This motif disappeared with increasing A-run length ($n = 207$ and 129, respectively). **i** An illustration of the inferred motifs in the context of replication. Source data are provided as a Source Data file.

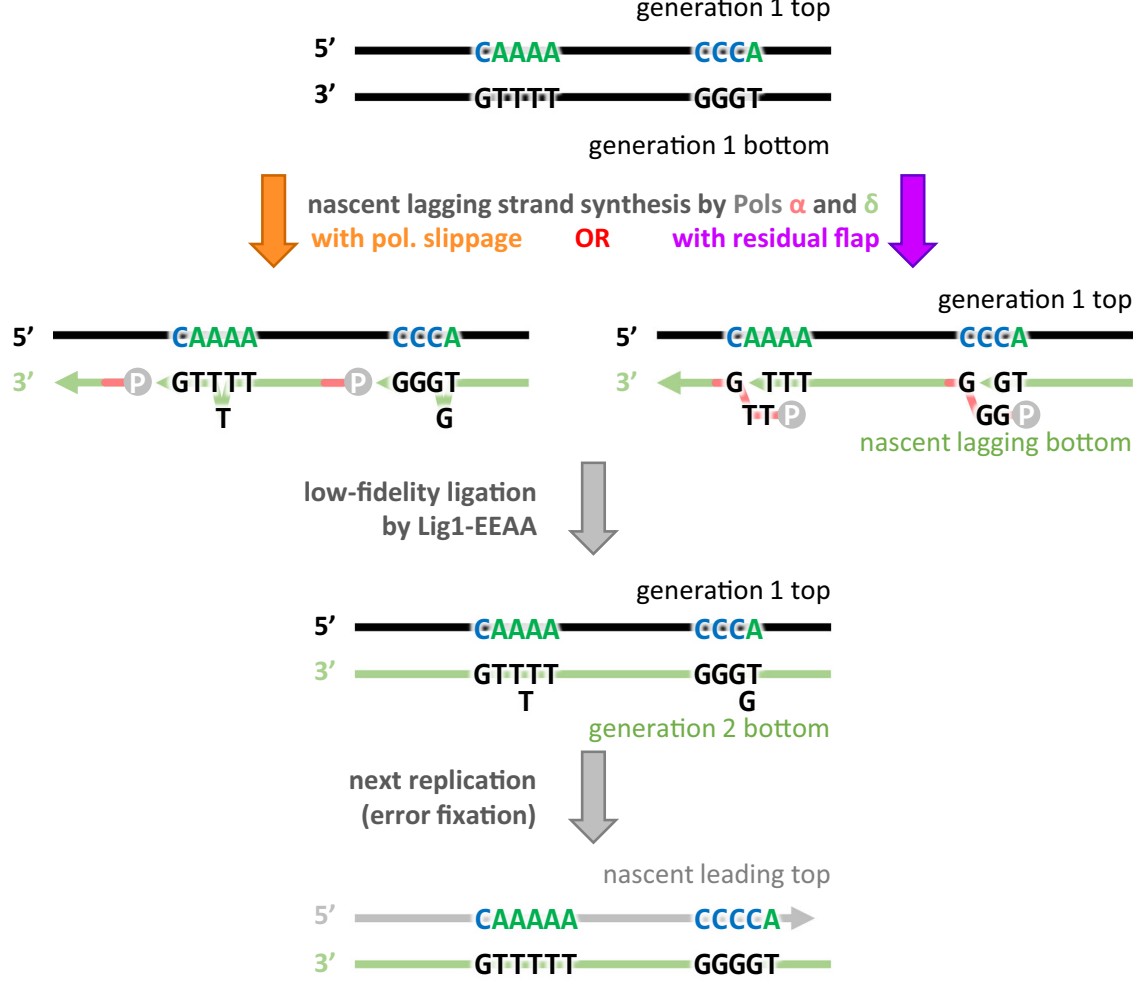

**Fig. 6 | Loss of high-fidelity ligation causes 1 bp lagging strand insertions in contexts with a super-motif of 5′-CₙAₘ-3′.** Proposed mechanisms for the creation of single-base insertions due to mutagenic ligation during lagging strand OFM. DNA strands are shown as lines with arrowheads indicating the direction of synthesis.

Bold letters indicate DNA bases involved with the motifs for single-base insertions due to the low-fidelity ligase mutant. Parental DNA strands are shown in black with letters colored to match motifs from Figs. 4, 5, nascent lagging strands in green with black letters, and terminal phosphates as gray circles.

of the number of mutations indicative of a given ploidy ($m_i$), the number of generations elapsed with that ploidy ($g_i$), the haploid genome size ($L$, in bp):

$$\mu_{\text{hap.}} = \frac{m_{\text{hom.}}}{L g_{\text{hap.}}};$$ (1)

and

$$\mu_{\text{dip.}} = \frac{m_{\text{het.}}}{2 L g_{\text{dip.}}}.$$ (2)

$L$, $m_{\text{hom.}}$ and $m_{\text{het.}}$ are directly measurable. The latter two are derived from the mutation totals, where

$$m_{\text{single}} = m_{\text{hom.}} + m_{\text{het.}}$$ (3)

arises when homozygous mutations are counted once ("*cdc9-EEAA msh2Δ*" in Supplementary Data 1) and

$$m_{\text{double}} = 2 m_{\text{hom.}} + m_{\text{het.}}$$ (4)

arises when they are counted twice ("*cdc9-EEAA msh2Δ* dubct" in Supplementary Data 1).

Rearranging Eq. 3 gives

$$m_{\text{hom.}} = m_{\text{double}} - m_{\text{single}}. \tag{5}$$

Substituting Eq. 5 into Eq. 4 and rearranging gives

$$m_{\text{het.}} = m_{\text{single}} - \left(m_{\text{double}} - m_{\text{single}}\right) = 2m_{\text{single}} - m_{\text{double}}. \tag{6}$$

Substituting Eqs. 5 and 6 into Eqs. 1 and 2, respectively, gives

$$\mu_{\text{hap.}} = \frac{m_{\text{double}} - m_{\text{single}}}{Lg_{\text{hap.}}} \tag{7}$$

and

$$\mu_{\text{dip.}} = \frac{2m_{\text{single}} - m_{\text{double}}}{2Lg_{\text{dip.}}}. \tag{8}$$

Elapsed generations are calculable assuming mutation rates or vice versa. We will use generation estimates to get the fraction of the experiment spent in a diploid state ($f_{\text{diploid}}$):

$$f_{\text{diploid}} = \frac{g_{\text{dip.}}}{g_{\text{dip.}} + g_{\text{hap.}}}. \tag{9}$$

Substituting in Eqs. 7 and 8 yields

$$f_{\text{diploid}} = \frac{\frac{2m_{\text{single}} - m_{\text{double}}}{2L\mu_{\text{dip.}}}}{\frac{2m_{\text{single}} - m_{\text{double}}}{2L\mu_{\text{dip.}}} + \frac{m_{\text{double}} - m_{\text{single}}}{L\mu_{\text{hap.}}}}, \tag{10}$$

which, assuming that the per base pair mutation rate is constant (i.e., $\mu_{hap.} = \mu_{dip.}$), simplifies to

$$f_{\text{diploid}} = \frac{2m_{\text{single}} - m_{\text{double}}}{2m_{\text{single}} - m_{\text{double}} + 2\left(m_{\text{double}} - m_{\text{single}}\right)} = \frac{2m_{\text{single}} - m_{\text{double}}}{m_{\text{double}}}$$
$$= \frac{2m_{\text{single}}}{m_{\text{double}}} - 1 \tag{11}$$

Supplementary Data 1 contains $f_{\text{diploid}}$ estimates for all *cdc9-EEAA msh2Δ* outgrowth samples ('outgrowths' tab).

### Statistical analysis

Unless otherwise noted, wherever mutation rates are compared between strains, significance was tested using the one-sided heteroscedastic Welch's *t*-test[37]. Degrees of freedom were approximated via the Welch–Satterthwaite equation[38]. Significance thresholds were set by applying the Šidák correction for multiple hypothesis testing[39] for a family-wise error rate of 0.05.

### Reporting summary

Further information on research design is available in the Nature Portfolio Reporting Summary linked to this article.

## Data availability

The DNA sequencing data generated in this study have been deposited in the Sequence Read Archive database under BioProject accession code PRJNA245050. Source data are provided with this paper.

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

## Acknowledgements

We thank the High Throughput Genomic Sequencing Facility at UNC Chapel Hill for sequencing. We are grateful to Kunkel lab members for helpful discussions, and Sarah Marks and Hunter Wilkins for critical reading of the manuscript. This work was supported by the intramural research program of the US National Institutes of Health (NIH), National Institute of Environmental Health Sciences (NIEHS) grants Z01ES065070 to T.A.K. and 1Z01ES102765 to R.S.W.

## Author contributions

Conceptualization and methodolgy, J.S.W., S.A.L., M.E.A., and T.A.K.; Investigation, J.S.W., S.A.L., and M.E.A.; Data analysis, J.S.W., S.A.L., A.B.B., P.P.T., R.S.W., and T.A.K.; Writing—original draft, J.S.W., S.A.L., and T.A.K.; Writing—review & editing, J.S.W., S.A.L., M.E.A., A.B.B., P.P.T., R.S.W., and T.A.K.; Supervision, T.A.K. and R.S.W.

## Funding

## Competing interests

The authors declare no competing interests.
