## [Peer Review File · Nature Communications]

High fidelity DNA ligation prevents single base insertions in the yeast genomeREVIEWER COMMENTS

Reviewer #1 (Remarks to the Author):

The present manuscript builds effectively on the mechanistic work detailed in a 2021 Nature Communications article by the same team. The earlier work demonstrated that a mutation in the metal-binding site of LigI leads to promiscuous ligation, impacting genomic stability. In the current work, the team has expanded on their findings, by doing whole genome sequencing to assess mutational hotspots in LigI mutant (*cdc9-EEAA*) *S. cerevisiae* genome.

Using prior knowledge of replication origins, the authors first show that there is a significant increase in genomic mutation rate in *cdc9-EEAA* strains compared to wild-type, which is substantially exacerbated in a *cdc9-EE/AA-msh2Δ* double mutant strain. Assessments of 1bp indels and 1bp insertions reveal that in the *cdc9-EEAA* mutant there is a high G.C insertion. This aligns with their structure model predictions, where a C or G nucleotide is expected to be accommodated in the metal binding site of the mutant *cdc9-EE/AA* allowing for low fidelity ligation. Examination of sequence context showed that the +G.C insertion rates were significantly high for most homopolymer lengths in strains containing either *cdc9EE/AA* or *cdc9EE/AA-msh2Δ* compared to the wild-type strain. Notably, G nucleotides were preferentially inserted on the lagging strand, and the presence of MMR (mismatch repair) reduced these insertions. Interestingly, the authors found no patterns for mutation hotspots in their analysis. Together, the manuscript provides convincing evidence that +1 mutations generated by the lagging strand polymerase, pol delta are efficiently cleaned up by *rad27* and the MMR pathway allowing for high-fidelity replication. However, DNA LigI is also an important contributor to replication fidelity by ensuring accurate ligation of nicked DNA. Promiscuous ligation of +1 insertions can introduce mutations throughout the genome, which is substantially increased in the absence of MMR.

Overall, this manuscript significantly enhances our understanding of factors involved in high-fidelity DNA replication. The data is well presented and the manuscript well written and easy to follow. Outstanding work and another impactful contribution to the replication field by this research team! I highly recommend publishing this manuscript without any suggested edits.

Reviewer #2 (Remarks to the Author):

Summary and general assessment:

Williams et al analyze the mutational landscape in *S. cerevisiae* when DNA ligase 1 (LIG1), involved in Okazaki fragment maturation (OFM), is altered in two conserved glutamate amino acids (Lig1-EEAA), rendering a mutator phenotype. In a previous study from the same group (Williams et al, Nat Commun 2021, ref. 12 in the current manuscript), the effect of Lig1-EEAA was assessed with a URA3 reporter gene; the authors have now extended this study to the entire genome, combining as needed Lig1-EEAA with a *msh2* mutant that loses the mismatch repair (MMR) pathway (Table 1 and Fig 1). The majority of mutations fixed by Lig1-EEAA in the absence of MMR are single base insertions in short mononucleotide repeats (Figure 2). Mutagenesis is biased towards the lagging strand, as it could be hypothesized for a ligation reaction that takes place mainly during OFM (Figure 3). In the case of G-C homopolymers, the absence of MMR favors G insertions over C insertions (Figures 3-4). This preference displays some sequence specificity (e.g. an A is usually present in the -1 position) and is attenuated in longer homopolymers (Figure 4). A similar analysis is conducted for A-T homopolymers, in which T insertions are generally favored over A insertions (Figure 5). Based on this molecular information, insertion hotspots are positioned along the different yeast chromosomes. These hotspots are located both within genes and in intergenic sequences, and no clear patterns are found in terms of transcriptional profile, replication timing or proximity to replication origins (Supplementary Figures).

The lab led by Dr Kunkel has made many important contributions to the molecular mechanisms that ensure fidelity during DNA replication. This expertise is evident throughout the manuscript and I have no technical criticisms to the experimental design or its execution. Regarding novelty, however, it is worth noting that their previous article on this topic (Williams et al, Nat Commun 2021) had analyzed in molecular and structural detail the effects of LIG1-EEAA and *msh2* in a

URA3 reporter gene. In this context, the current study does not provide much conceptual advance, besides the evident advance of expanding the mutational analysis to the genome-wide level. Having said this, the identification of mutational hotspots due to the combination of polymerase slippage, faulty ligation and loss of MMR is potentially valuable to the field, and I think the manuscript could be made stronger if the authors went a bit deeper into the characterization of the hotspots that they have already identified, using computational genomics tools (e.g. test the possible correlations with epigenetic marks).

Minor points:

Lines 67-70, the sentence is repetitive.

l. 127. According to the values shown, the increase in genomic mutation rate should be 950-fold, not 95-fold.

l. 141. I believe the authors refer to Table 1b, not Table 1a.

l. 209. Duplicated word ("in in")

l. 394. "Stand" should read "strand"

l. 415 and Figure 4 e-h. "Green arrows denote the direction of primer strand synthesis" seems misplaced (should be in Fig 4i legend).

Reviewer #3 (Remarks to the Author):

In 2021 Williams and colleagues published a paper in Nature Communications that elegantly combined yeast genetics, mutagenesis reporter assays, biochemistry and structural biology to show that eukaryotic DNA ligase I (Cdc9 in yeast) has a high-affinity metal binding site that safeguards against 1bp insertions. Mutation of this binding site creates space for the bulge that forms when an additional base is present in the newly synthesised strand. Therefore, whereas wildtype DNA ligase I will not normally ligate two fragments in which the 5' fragment has an insertion in the top strand (or will only do so with very low efficiency), the mutant DNA ligase I is able to. In vivo, such insertion mutations are likely to occur through DNA polymerase slippage and be particularly relevant during strand displacement synthesis by Pol delta during Okazaki fragment maturation. This earlier work established that DNA ligase I plays an important role in preventing insertion mutagenesis. In addition, it showed that MMR and FEN1/Rad27 also protect against such mutations, at least in DNA ligase I mutant yeast.

In this manuscript Williams and colleagues extend these earlier findings beyond the URA3 reporter gene, showing that: 1) the increase in 1bp insertions in DNA ligase I mutant yeast occurs genome-wide, 2) the same mutations are further increased when combined with MMR deficiency, 3) they are enriched in specific sequence contexts, and 4) they almost certainly occur during lagging strand synthesis. Indel signatures have been defined in cancer, but for many of these the underlying mechanisms remain to be defined. Therefore, defining indel mutagenesis mechanisms in model organisms is valuable. Important unanswered questions raised by this work are: how likely are mutations involving this mechanism to occur in cells expressing wildtype DNA ligase I; and what other processes (e.g. those related to other OFM proteins) promote this type of insertion mutagenesis? Future answers to these questions should clarify how important this process may be for human mutagenesis and cancer. Although no direct link has yet been made here to one of the COSMIC signatures, it may aid others in their investigations.

Altogether this is a well conducted study that provides further insight into a novel mutagenic process first described by the same authors in 2021. It provides useful information for those studying replication and indel mutation signature.

I have a number of questions that I'd like to see addressed:

- I'm confused by the ploidy of the *cdc9/msh2* double mutant strain used in the genome-wide mutation experiment. In the methods there is mention of acquired diploidy for all double mutant strains; in Supplementary Fig 1 all strains (including those for 0 generations) are indicated to have a ploidy of 2, and Table S1b does not show details for a double mutant diploid strain. This suggests that the intention was to perform this experiment with a haploid double mutant strain, which is surprising considering the reduced impact of purifying selection when using diploid strains, and the comparison with wildtype and single mutant diploid strains. Why was this decision made? If diploidy was indeed acquired, how would this have happened? Also, it should be possible to work out whether the double mutant strain was diploid at the start of the experiment (assuming stocks were kept of generation 0 isolates). If this was indeed the case, there is no complication with regards to the possibility of acquired diploidy during the experiment. If it was haploid to start with however, it could potentially underestimate the mutation rate (due to stronger purifying selection up to the point where a diploid state was acquired) and should at least be mentioned. I do not expect any of this to impact on the validity of the findings, but some clarity would be helpful.
- Why is the repeat length peak for insertions different for T/A insertion than for G/C insertions? Is there a structural explanation for this (considering the structure presented in the 2021 paper which suggests that the bulge is accommodated in a specific alignment register)? Also, do the authors have a (structural) explanation or hypothesis for why there is a preference for G>C>T>A?
- In the 2021 paper it is mentioned that "we are currently testing whether the 3'-exonuclease activity of Pol delta is capable of proofreading such replication errors". If those experiments gave a clear answer, it would be a valuable addition to this manuscript, as it is relevant for a potential link between this mechanism of insertion mutagenesis and cancer mutational signatures, with mutations in *POLD1* that affect the 3'-exo activity occurring in certain cancers.
- If the proposed model is correct, mutations would be expected to be enriched at OF junctions, which are enriched at nucleosome binding sites and certain transcription factor binding sites (PMID: 22419157, 25624100). Indeed, in the 2021 paper hotspots in the *URA3* reporter were linked to nucleosome binding due to the distance between them. Does this hold up with the genome-wide data? Although there is no obvious pattern apparent in the data in Fig S5, better analyses could be done using genome-wide nucleosome binding data.
- Personally, I would find things easier to follow (and perhaps others would too?) if in text and figures (Fig 3d,e; Fig 4,5; Fig S3,4) the authors would present things from the perspective of the top strand being synthesised by lagging strand polymerases, so that the focus is on the newly synthesised strand (the likely source of the insertions) and the motifs relate to this strand as well (so that addition of G is most likely in an RTGGG context). If there is a specific reason to focus on the template strand instead, can the authors please explain why?
- Fig 4 and 5. For insertion sequence contexts, it is unlikely that there is a strong sequence context genome-wide for runs of G/Cs or T/As. However, to exclude that sequence bias impacts on the identified motifs it would be good to show logos for genome-wide flanking sequences for 3G/C, 4G/C etc. For all sequence logos, it would be helpful to put the most common base on top (as is custom) and for Fig 5, the y-axis should go to 2 bit for fair comparison with Fig 4 (as it currently stands, it gives the impression of a stronger motif for T insertions than there actually is).
- Both *Rad27* and MMR deficiency increase the *cdc9-EEAA* driven insertion mutations. This is explained by a role for *Rad27* in OF flap cleavage and a role for MMR in repairing +1bp mismatches, and it is acknowledged in the discussion that MMR machinery may also have a role in OF flap cleavage. However, it is not clear to me how loss of *Rad27* (or MMR) and the impact of this on OF flap cleavage would lead to insertion mutagenesis caused by *cdc9-EEAA* (see below; a cartoon model would be helpful). Instead, is it not more plausible that the increase in the *Rad27* null *cdc9-EEAA* strain is explained by *Rad27* playing a potential role in MMR, particularly when this occurs at the junction between OFs? (see PMID: 34552065)
- In the discussion, the authors talk about three possible models about the insertion mechanism, but they only present one model in Fig 5. It would be helpful to better understand the alternative models if cartoons were added for these (perhaps in supplementary) for possibilities 2 and 3. For possibility 2, is this consistent with the ligase structure with the bulge-containing substrate? I don't understand possibility 3; why would mutations be lagging strand enriched in that case?

Minor:

- Line 65-67: the relevance of *CDC9* overexpression on *Rad27*-dependent OFM and duplication/deletion mutations is not clear to me in relation to the work presented here

- Line 111-112: please explain where the 2 to 7 fold preference for G over C comes from in relation to the experiments with URA3-OR1 and OR2 reporters
- Fig 1a, it would be helpful to indicate a vertical dotted line for ratio 1
- In Fig 3 the detection limits are different for different strains. I presume this is because this is because of different numbers of total generations? It would be helpful if the method of calculating the detection limits is explained.
- Please explain the calculation in the legend of Fig 3e. My calculations suggest it should be 7.7 ($1/(0.881-0.751)$), not 7.4.
- What does the heatmap "significance threshold" mean? E.g. Fig S2
- Fig S3: what does 64:41 bias mean? What does 50% inter-ACS mean?
- Line 227: I'm not convinced it makes sense to combine motifs for G/C and A/T insertions, but if doing so n and m should be defined.
- Line 245-246: More mutations than expected in intergenic regions. Is this due to purifying selection for mutations occurring in coding sequences or because there is less well defined nucleosome binding sites in these regions?

REVIEWERS' COMMENTS

Reviewer #1:

The present manuscript builds effectively on the mechanistic work detailed in a 2021 Nature Communications article by the same team. The earlier work demonstrated that a mutation in the metal-binding site of LigI leads to promiscuous ligation, impacting genomic stability. In the current work, the team has expanded on their findings, by doing whole genome sequencing to assess mutational hotspots in LigI mutant (cdc9-EEAA) <S. cerevisiae> genome.

Using prior knowledge of replication origins, the authors first show that there is a significant increase in genomic mutation rate in cdc9-EEAA strains compared to wild-type, which is substantially exacerbated in a cdc9-EE/AA-msh2Δ double mutant strain. Assessments of 1bp indels and 1bp insertions reveal that in the cdc9-EEAA mutant there is a high G.C insertion. This aligns with their structure model predictions, where a C or G nucleotide is expected to be accommodated in the metal binding site of the mutant cdc9-EE/AA allowing for low fidelity ligation. Examination of sequence context showed that the +G.C insertion rates were significantly high for most homopolymer lengths in strains containing either cdc9EE/AA or cdc9EE/AA-msh2Δ compared to the wild-type strain. Notably, G nucleotides were preferentially inserted on the lagging strand, and the presence of MMR (mismatch repair) reduced these insertions. Interestingly, the authors found no patterns for mutation hotspots in their analysis. Together, the manuscript provides convincing evidence that +1 mutations generated by the lagging strand polymerase, pol delta are efficiently cleaned up by rad27 and the MMR pathway allowing for high-fidelity replication. However, DNA LigI is also an important contributor to replication fidelity by ensuring accurate ligation of nicked DNA. Promiscuous ligation of +1 insertions can introduce mutations throughout the genome, which is substantially increased in the absence of MMR.

Overall, this manuscript significantly enhances our understanding of factors involved in high-fidelity DNA replication. The data is well presented and the manuscript well written and easy to follow. Outstanding work and another impactful contribution to the replication field by this research team! I highly recommend publishing this manuscript without any suggested edits.

Response: We thank the reviewer for their positive comments on the manuscript.

Reviewer #2:

Summary and general assessment:

Williams et al analyze the mutational landscape in S. cerevisiae when DNA ligase 1 (LIG1), involved in Okazaki fragment maturation (OFM), is altered in two conserved glutamate amino acids (Lig1-EEAA), rendering a mutator phenotype. In a previous study from the same group (Williams et al, Nat Commun 2021, ref. 12 in the current manuscript), the effect of Lig1-EEAA was assessed with a URA3 reporter gene; the authors have now extended this study to the entire genome, combining as needed Lig1-EEAA with a msh2 mutant that loses the mismatch repair (MMR) pathway (Table 1 and Fig 1). The majority of mutations fixed by Lig1-EEAA in the absence of MMR are single base insertions in short mononucleotide repeats (Figure 2). Mutagenesis is biased towards the lagging strand, as it could be hypothesized for a ligation reaction that takes place mainly during OFM (Figure 3). In the case of G-C homopolymers, the

absence of MMR favors G insertions over C insertions (Figures 3-4). This preference displays some sequence specificity (e.g. an A is usually present in the -1 position) and is attenuated in longer homopolymers (Figure 4). A similar analysis is conducted for A-T homopolymers, in which T insertions are generally favored over A insertions (Figure 5). Based on this molecular information, insertion hotspots are positioned along the different yeast chromosomes. These hotspots are located both within genes and in intergenic sequences, and no clear patterns are found in terms of transcriptional profile, replication timing or proximity to replication origins (Supplementary Figures).

*The lab led by Dr Kunkel has made many important contributions to the molecular mechanisms that ensure fidelity during DNA replication. This expertise is evident throughout the manuscript and I have no technical criticisms to the experimental design or its execution. Regarding novelty, however, it is worth noting that their previous article on this topic (Williams et al, Nat Commun 2021) had analyzed in molecular and structural detail the effects of *LIG1-EEAA* and *msh2* in a *URA3* reporter gene. In this context, the current study does not provide much conceptual advance, besides the evident advance of expanding the mutational analysis to the genome-wide level. Having said this, the identification of mutational hotspots due to the combination of polymerase slippage, faulty ligation and loss of MMR is potentially valuable to the field, and I think the manuscript could be made stronger if the authors went a bit deeper into the characterization of the hotspots that they have already identified, using computational genomics tools (e.g. test the possible correlations with epigenetic marks).*

Response: We thank the reviewer for their positive comments on the manuscript. We appreciate the suggestion to strengthen the manuscript by going deeper into hotspot characterization.

In future studies, we agree that it will be important to study the effect of epigenetic regulation that includes histone modifications and chromatin organization on ligation fidelity using the *cdc9-EEAA* mutant. This will be especially informative in mammalian cells with respect to DNA methylation, and as these marks are heritable yet dynamic during development. However, unlike nucleosome positions, epigenetic markers have not been mapped in our yeast strain background.

Minor points:

Lines 67-70, the sentence is repetitive.

Response: This has been corrected and changed to: ‘Genome stability requires efficient and accurate ligation of the abundant DNA nicks produced during OFM which are generated at replication origins and about every 250 base pairs during lagging strand replication.’

l. 127. According to the values shown, the increase in genomic mutation rate should be 950-fold, not 95-fold.

Response: This has been corrected and changed to 950-fold.

l. 141. I believe the authors refer to Table 1b, not Table 1a.

Response: The reviewer is correct and we changed this to reference Table 1b.

l. 209. Duplicated word (“in in”)

Response: This has been corrected.

l. 394. “Stand” should read “strand”

Response: This has been corrected.

l. 415 and Figure 4 e-h. “Green arrows denote the direction of primer strand synthesis” seems misplaced (should be in Fig 4i legend).

Response: This has been corrected and moved to the Fig. 4i legend.

Reviewer #3:

In 2021 Williams and colleagues published a paper in Nature Communications that elegantly combined yeast genetics, mutagenesis reporter assays, biochemistry and structural biology to show that eukaryotic DNA ligase I (Cdc9 in yeast) has a high-affinity metal binding site that safeguards against 1bp insertions. Mutation of this binding site creates space for the bulge that forms when an additional base is present in the newly synthesized strand. Therefore, whereas wildtype DNA ligase I will not normally ligate two fragments in which the 5' fragment has an insertion in the top strand (or will only do so with very low efficiency), the mutant DNA ligase I is able to. In vivo, such insertion mutations are likely to occur through DNA polymerase slippage and be particularly relevant during strand displacement synthesis by Pol delta during Okazaki fragment maturation. This earlier work established that DNA ligase I plays an important role in preventing insertion mutagenesis. In addition, it showed that MMR and FEN1/Rad27 also protect against such mutations, at least in DNA ligase I mutant yeast.

In this manuscript Williams and colleagues extend these earlier findings beyond the URA3 reporter gene, showing that: 1) the increase in 1bp insertions in DNA ligase I mutant yeast occurs genome-wide, 2) the same mutations are further increased when combined with MMR deficiency, 3) they are enriched in specific sequence contexts, and 4) they almost certainly occur during lagging strand synthesis. Indel signatures have been defined in cancer, but for many of these the underlying mechanisms remain to be defined. Therefore, defining indel mutagenesis mechanisms in model organisms is valuable. Important unanswered questions raised by this work are: how likely are mutations involving this mechanism to occur in cells expressing wildtype DNA ligase I; and what other processes (e.g. those related to other OFM proteins) promote this type of insertion mutagenesis? Future answers to these questions should clarify how important this process may be for human mutagenesis and cancer. Although no direct link has yet been made here to one of the COSMIC signatures, it may aid others in their investigations.

Altogether this is a well conducted study that provides further insight into a novel mutagenic process first described by the same authors in 2021. It provides useful information for those studying replication and indel mutation signature.

Response: We thank the reviewer for their positive comments.

I have a number of questions that I'd like to see addressed:

*- I'm confused by the ploidy of the *cdc9/msh2* double mutant strain used in the genome-wide mutation experiment. In the methods there is mention of acquired diploidy for all double mutant strains; in Supplementary Fig 1 all strains (including those for 0 generations) are indicated to have a ploidy of 2, and Table S1b does not show details for a double mutant diploid strain. This suggests that the intention was to perform this experiment with a haploid double mutant strain, which is surprising considering the reduced impact of purifying selection when using diploid strains, and the comparison with wildtype and single mutant diploid strains. Why was this decision made? If diploidy was indeed acquired, how would this have happened? Also, it should be possible to work out whether the double mutant strain was diploid at the start of the experiment (assuming stocks were kept of generation 0 isolates). If this was indeed the case, there is no complication with regards to the possibility of acquired diploidy during the experiment. If it was haploid to start with however, it could potentially underestimate the mutation rate (due to stronger purifying selection up to the point where a diploid state was acquired) and should at least be mentioned. I do not expect any of this to impact on the validity of the findings, but some clarity would be helpful.*

Response: We apologize for the confusion. Our initial intention was to use diploid strains homozygous for *cdc9-EEAA* and *msh2Δ*. However, we had significant difficulty constructing this strain using multiple independent approaches, and we hypothesize this may relate to instability caused by aneuploidy that may be causing unequal chromosome segregation during meiosis. Spontaneous changes in ploidy are common in yeast, with the two main diploidization mechanisms found to be whole-genome duplication and spontaneous mating-type switching, even with heterothallic strains (PMID: 29502947). Various selective pressures, including high mutation loads, promote diploidization. This is not our first strain wherein all lineages changed ploidy, with the most extreme example going from diploid to a mix of triploid and tetraploid in all lineages before the start of mutation accumulation. Mutation rates in this manuscript are far lower, but haploids are both more affected by individual mutations and more susceptible to spontaneous ploidy changes.

The reviewer is correct that we know which isolates were diploid prior to the experiment, though not because of the generation 0 samples. We can show that rates and selection differed little in haploid versus diploid states and can estimate the generation in which diploidization occurred. These estimates have been added to Supplementary Data File 1.

Here is the methodology behind these assertions:

Strains that are sufficiently heterozygous reveal their ploidy in allelic fraction plots. All *cdc9-EEAA msh2Δ* lines had sufficient heterozygosity by the end of the experiment, but none had sufficient heterozygosity in generation 0. Instead, we used diploid settings in the Muvet mutation calling pipeline to cause mutations accumulated before and after diploidization to be called as

homozygous and heterozygous, respectively. If the vast majority of “homozygous” mutations are due to mutations prior to diploidization AND there is little underestimation of rates in the haploid state, then the “true” rate would be approximately the heterozygous rate plus twice the homozygous rate. We call this the “double counting” rate. If these assumptions are met, then the inter-isolate relative variance in double counting rates should be (A) lower than for the relative variance in heterozygous rates, and (B) comparable to or lower than the variance in other strains with similar rates. We have conducted mutation accumulations in 92 *S. cerevisiae* strains. The mean relative variance is 31%. The mean relative variance in other strains with rates between 100 and 1000 Gbp⁻¹ generation⁻¹ (16 strains) is 26%. The relative variance in the *cdc9-EEAA msh2Δ* heterozygous and “double counting” rates are 31% and 13%, respectively. Thus the “double counting” rates pass tests A and B, suggesting little selection in the haploid state and relatively constant rates per Gbp regardless of ploidy. This lets us estimate that diploidization occurred within the first 15% of the experiment in 21 isolates, within the last 15% of the experiment in 12 isolates, and in between for 6 isolates.

- Why is the repeat length peak for insertions different for T/A insertion than for G/C insertions? Is there a structural explanation for this (considering the structure presented in the 2021 paper which suggests that the bulge is accommodated in a specific alignment register)? Also, do the authors have a (structural) explanation or hypothesis for why there is a preference for G>C>T>A?

Response: The data indicate an order of G>T>C>A and could be dictated by the fidelity of the DNA polymerase during synthesis, the fidelity of DNA ligase during nick sealing, or some combination of both. Assessing this is of great interest to us but will require detailed biochemical analysis beyond the scope of the current work.

- In the 2021 paper it is mentioned that “we are currently testing whether the 3’-exonuclease activity of Pol delta is capable of proofreading such replication errors”. If those experiments gave a clear answer, it would be a valuable addition to this manuscript, as it is relevant for a potential link between this mechanism of insertion mutagenesis and cancer mutational signatures, with mutations in POLDI that affect the 3’-exo activity occurring in certain cancers.

Response: We agree that these experiments have value and are currently testing the possibility that Pol δ proofreads insertion errors by measuring *URA3* mutation rates and determining mutation spectra in a Pol δ proofreading-deficient mutant (*pol3-5DV*) expressing the low fidelity *cdc9-EEAA* allele and comparing them to wt and the *pol3-5DV* and *cdc9-EEAA* single mutant strains. These experiments were initiated by a previous member of the lab who moved onto a new position before the project was completed, and our plan is to complete this mutational reporter analysis and also perform whole genome sequencing on these strains. We are also testing a role for Pol ε proofreading of 1 bp insertion errors during replication using a *pol2-4* exonuclease-deficient version of Pol ε.

- If the proposed model is correct, mutations would be expected to be enriched at OF junctions, which are enriched at nucleosome bindings sites and certain transcription factor bindings sites (PMID: 22419157, 25624100). Indeed, in the 2021 paper hotspots in the URA3 reporter were linked to nucleosome binding due to the distance between them. Does this hold up with the

genome-wide data? Although there is no obvious pattern apparent in the data in Fig S5, better analyses could be done using genome-wide nucleosome binding data.

Response: The reviewer is correct on all points, which is why we did analyze rates relative to both genome-wide nucleosome binding positions and origins of replication. However, we elected not to show them because, as with ORFs and replication timing in the existing figure, all trends were either overwhelmed by other MMR- mutation sources (i.e. substitutions and non-G+ indels (Lujan, 2014)) or were explainable using only G-run density. Given that the vast majority of well-ordered nucleosomes are already in the figure (flanking the UTRs), we judged these analyses to be redundant.

In the original draft we point out that “we observed a significant number of +1 mutations ... [which] ... likely occur at a small subset of the total ligation events”, i.e. rare events only in those locations where the target sequence coexists with Okazaki fragment termini. Evidently the target sequence density dominates the OF terminal density patterns at this resolution.

- Personally, I would find things easier to follow (and perhaps others would too?) if in text and figures (Fig 3d,e; Fig 4,5; Fig S3,4) the authors would present things from the perspective of the top strand being synthesised by lagging strand polymerases, so that the focus is on the newly synthesised strand (the likely source of the insertions) and the motifs relate to this strand as well (so that addition of G is most likely in an RTGGG context). If there is a specific reason to focus on the template strand instead, can the authors please explain why?

Response: The converging forks depicted in Figs. 3a-b, 4i, and 5i are presented as per convention, with the reference strand presented as top strand in the 5'-to-3' orientation. This means that the nascent lagging strand will be the bottom strand to the right of origins. This is the first representation that the reader will see, so we chose to unify the sequence logos with this depiction, which then determined the strand used when discussing contexts.

- Fig 4 and 5. For insertion sequence contexts, it is unlikely that there is a strong sequence context genome-wide for runs of G/Cs or T/As. However, to exclude that sequence bias impacts on the identified motifs it would be good to show logos for genome-wide flanking sequences for 3G/C, 4G/C etc. For all sequence logos, it would be helpful to put the most common base on top (as is custom) and for Fig 5, the y-axis should go to 2 bit for fair comparison with Fig 4 (as it currently stands, it gives the impression of a stronger motif for T insertions than there actually is).

Response: Once you account for which polymerase replicates which strand, and therefore the replication direction relative to each run, there are slight flanking sequence biases. However, these are small enough to not change any conclusions of this study.

Rather than adding 5 new logos, we now include the following Supplementary Table 5: Frequencies of A at the +1 positions flanking G runs and C at the -1 positions flanking T runs, given 38% GC content in yeast and restricted to zones where polymerase usage is known in > 90% of replications:

Homopolymer	3 Cs	4 Cs	5 Cs	4 As	5 As
Expected	0.38	0.38	0.38	0.28	0.28
All runs	0.47	0.51	0.51	0.33	0.32
Runs with insertion	0.98	0.64	0.68	0.64	0.61

Our sequence logos use a custom tool that automatically accounts for global GC fractions and the exclusion of certain bases from the flank based on the run definition (i.e. G can't flank a G-run, A can't precede a CA dinucleotide tract, etc.). This tool does not allow for reordering of letter stacking in the logos.

- Both Rad27 and MMR deficiency increase the cdc9-EEAA driven insertion mutations. This is explained by a role for Rad27 in OF flap cleavage and a role for MMR in repairing +1bp mismatches, and it is acknowledged in the discussion that MMR machinery may also have a role in OF flap cleavage. However, it is not clear to me how loss of Rad27 (or MMR) and the impact of this on OF flap cleavage would lead to insertion mutagenesis caused by cdc9-EEAA (see below; a cartoon model would be helpful). Instead, is it not more plausible that the increase in the Rad27 null cdc9-EEAA strain is explained by Rad27 playing a potential role in MMR, particularly when this occurs at the junction between OFs? (see PMID: 34552065)

Response: Whole genome *cdc9-EEAA msh2Δ* indel rates exceed *msh2Δ* indel rates. This means that many indels must proceed through an MMR-independent pathway. Rate ratios suggest that this includes most deletions and the vast majority of insertions. The same is true in whole genome *rad27Δ* and *cdc9-EEAA rad27Δ* indel rates exceed *msh2Δ* indel rates (unpublished data; manuscripts in progress).

- In the discussion, the authors talk about three possible models about the insertion mechanism, but they only present one model in Fig 5. It would be helpful to better understand the alternative models if cartoons were added for these (perhaps in supplementary) for possibilities 2 and 3. For possibility 2, is this consistent with the ligase structure with the bulge-containing substrate? I don't understand possibility 3; why would mutations be laggings strand enriched in that case?

Response: Upon careful consideration, we have decided to remove any discussion of possibility 3. We thank the reviewer for the suggestion to modify Fig. 6 and illustrations for models 1 and 2 are now included and shown below:

Minor:

- Line 65-67: the relevance of CDC9 overexpression on Rad27-dependent OFM and duplication/deletion mutations is not clear to me in relation to the work presented here.

Response: This sentence has been removed.

- Line 111-112: please explain where the 2 to 7 fold preference for G over C comes from in relation to the experiments with URA3-OR1 and OR2 reporters.

Response: The 2-7 fold increase comes from dividing the +1 rate observed using the *URA3-OR2* reporter by the rate observed using *URA3-OR1* at the hotspot positions (344, 564 and 612) in the *cdc9-EEAA msh2Δ* strain. When the *URA3* reporter is in OR2, the newly synthesized lagging strand would contain a +G at each of the hotspots, whereas it would contain a +C in *URA3-OR1*. We now expanded the text to include the following: ‘Thus, in accordance with the model for accommodation of an extra nucleotide due to structural plasticity imparted by the cavity created by mutating the high-fidelity metal-binding site in Cdc9, the bulged nucleotide could theoretically be either a C or a G¹², but with a two to seven-fold preference for G when comparing the +1 rates at the hotspots in *URA3-OR1* versus *-OR2* for the *cdc9-EEAA msh2Δ* strain (Supplementary Table 3), and with a preference for G•C runs flanked by a T•A base pair (Supplementary Fig. 1).

- Fig 1a, it would be helpful to indicate a vertical dotted line for ratio 1

Response: A vertical dotted line for a ratio of 1 is now included in Fig. 1a and described in the legend.

- In Fig 3 the detection limits are different for different strains. I presume this is because this is because of different numbers of total generations? It would be helpful if the method of calculating the detection limits is explained.

Response: We believe that the reviewer is referring to Fig. 2, and they are correct that the limits differ by the number of isolates/generations of mutation accumulation. The first paragraph of the ‘‘Sequencing data analysis and genomic mutation rates’’ section of the Methods states:

‘‘Mutation rates were calculated as previously reported (37). Briefly, for a given mutation type and context, the mutation rate is equal to the mutation count divided by the number of elapsed cellular generations and the number of base pairs where such a mutation could occur.’’

The legend for Fig. 2 states:

‘‘Detection limits are the rates that would be implied by one observed insertion with a given base content and homopolymer length.’’

We have added the following to the Fig. 2 legend on page 17.

“The minimum detectable rate for a given context, the detection limit, is the rate that would be calculated if one mutation were to be observed in one instance of that context.”

- Please explain the calculation in the legend of Fig 3e. My calculations suggest it should be 7.7 (1/(0.881-0.751), not 7.4.

Response: If the y-axis represents the fraction of G insertions, then the fraction of G insertions at $f_{BS\alpha\delta} = 0$ will be the y-intercept and the fraction of C insertions will be 1 minus the y-intercept, so the ratio is $0.881/(1-0.881) = 0.74$.

- What does the heatmap “significance threshold” mean? E.g. Fig S2

Response: As stated in the legends for Fig. 2g and Supplementary Figure 2, “Bins with counts exceeding the significance threshold have a black border (Šidák correction; each bin counts as a hypothesis tested; family-wise error rate = 0.05).” This is how many observations would be required to exceed expectations due to random chance given the number of bins (~1200), the total number of observations, and a binomial distribution of counts per bin. In order to avoid false positives, each bin must be considered a separate test of the null hypothesis (that the count does not differ from expectations). We determine the threshold using Šidák correction and a family-wise error rate of 0.05. This is a very stringent test, which made the hotspots in Fig. 2 all the more surprising.

- Fig S3: what does 64:41 bias mean? What does 50% inter-ACS mean?

Response: We apologize for the confusion. This is shorthand used in the lab and has been removed from the Figure. Further, the first line of the Supplementary Fig. 3 legend now reads “As per Fig. 4e but drawing from only the first and last 50% of each inter-origin space, a sequence logo illustrates the motif for G insertions in the *cdc9-EEAA* strain.”

- Line 227: I’m not convinced it makes sense to combine motifs for G/C and A/T insertions, but if doing so n and m should be defined.

Response: Thank you for your feedback. We have changed this sentence to read “Combining the *cdc9-EEAA msh2Δ* motifs lead to a model in which single base lagging strand insertions occur during OFM in contexts with a template super-motif of 5'-C_nA_m-3'. The highest insertion rates were found in contexts where (n,m) = (3 to 5,1) or (1,4 to 5).”

- Line 245-246: More mutations than expected in intergenic regions. Is this due to purifying selection for mutations occurring in coding sequences or because there is less well defined nucleosome binding sites in these regions?

Response: It is possible that the slight overrepresentation of hotspots around genes can be attributed to purifying selection. However, we have evidence that mutation rates are essentially constant across haploid and diploid states. Since haploids should be more susceptible to selection, this suggests that there is not much purifying selection during our accumulation experiments. We are now attempting a more sensitive experiment to investigate differences in

selection biases between indels of $3n$ bp and those of $f(n) = 3n \pm 1$ bp (e.g. single-base insertions: $f(0) = 3 \times 0 + 1 = 1$). We hope to relate the results to published strains, including *cdc9-EEAA msh2Δ*.

REVIEWERS' COMMENTS

Reviewer #3 (Remarks to the Author):

The authors have answered most of my questions and made changes to the manuscript to clarify things to the reader.

I have no concerns about the scientific validity of the findings presented. However, I would suggest a small number of additional minor changes that I feel would be helpful to the reader.

- In their rebuttal the authors provide a comprehensive explanation about the acquired diploidy and how they dealt with this in their experiment. And in the new version of the methods they provide in-depth technical information. This is a useful addition, but it would be helpful if they could also add a sentence before "All cdc9-EEAA msh2 Δ lines were found to have acquired diploidy either before or during the mutation accumulating experiment" to explain that the experiment was originally started with a haploid strain because diploid generation was not successful, "however all cdc9-EEAA msh2 Δ lines were found to have acquired diploidy", to explain to readers of the manuscript why a diploid strain was not used to start with.

- I appreciate the addition of Suppl Table 5 to show that the sequence motif for 1bp G/T insertions is not due to genome-wide bias of the sequence context around G or T repeats. However, if people don't read the rebuttal it is not easy to follow. The legend for this table could be used to explain better why this analysis was performed and what it means. Also, it should be possible to perform a Chi Square test and provide p-values to show that the observed motif for the number of observations is significant relative to random expectation.

- The first part of my original question "Why is the repeat length peak for insertions different for T/A insertion than for G/C insertions? Is there a structural explanation for this (considering the structure presented in the 2021 paper which suggests that the bulge is accommodated in a specific alignment register)? Also, do the authors have a (structural) explanation or hypothesis for why there is a preference for G>C>T>A?" was not responded to, and I understand and except that the second part will be addressed in future work. This is not essential, but perhaps these could be briefly commented on (e.g. in the discussion) as outstanding questions?

REVIEWERS' COMMENTS

Reviewer #3 (Remarks to the Author):

The authors have answered most of my questions and made changes to the manuscript to clarify things to the reader.

I have no concerns about the scientific validity of the findings presented. However, I would suggest a small number of additional minor changes that I feel would be helpful to the reader.

Response: We thank the reviewer for their positive comments and have incorporated the additional minor changes as suggested below.

*- In their rebuttal the authors provide a comprehensive explanation about the acquired diploidy and how they dealt with this in their experiment. And in the new version of the methods they provide in-depth technical information. This is a useful addition, but it would be helpful if they could also add a sentence before “All *cdc9-EEAA msh2Δ* lines were found to have acquired diploidy either before or during the mutation accumulating experiment” to explain that the experiment was originally started with a haploid strain because diploid generation was not successful, “however all *cdc9-EEAA msh2Δ* lines were found to have acquired diploidy”, to explain to readers of the manuscript why a diploid strain was not used to start with.*

Response: Thank you for this suggestion. We have now included the following sentence on page 16 to clarify the issue of ploidy in the passaging experiment. “Diploid strain construction using multiple independent approaches was unsuccessful, so haploid strains were constructed for passaging and mutation accumulation. However, all *cdc9-EEAA msh2Δ* lines were found to have acquired diploidy either before or during the mutation accumulation experiment”.

- I appreciate the addition of Suppl Table 5 to show that the sequence motif for 1bp G/T insertions is not due to genome-wide bias of the sequence context around G or T repeats. However, if people don't read the rebuttal it is not easy to follow. The legend for this table could be used to explain better why this analysis was performed and what it means. Also, it should be possible to perform a Chi Square test and provide p-values to show that the observed motif for the number of observations is significant relative to random expectation.

Response: Supplementary Table 5 has been expanded to include counts and fractions of insertions matching the motif for each homopolymer length and Chi-squared test p-values showing the deviation from expected fractions.

all C insertions					
$p < 10^{-150}$	-2 C	-1 R	CCC	+1 A	+2 Y
N (insert count)	1010	2333	2857	2802	2654
f_{expected}	0.198	0.624		0.473	0.527
f_{observed}	0.354	0.817		0.981	0.929
$p = 2.3 \times 10^{-21}$	-2 C	-1 R	CCCC	+1 A	+2 Y
N	330	1079	1529	1005	888

f_{expected}	0.194	0.617		0.506	0.531
f_{observed}	0.216	0.706		0.657	0.581
$p = 1.0 \times 10^{-4}$	-2 C	-1 R	CCCCC	+1 A	+2 Y
N	177	417	646	386	376
f_{expected}	0.231	0.637		0.509	0.514
f_{observed}	0.274	0.646		0.598	0.582
$p = 2.2 \times 10^{-2}$	-2 C	-1 R	CCCCCC	+1 A	+2 Y
N	44	96	184	94	102
f_{expected}	0.325	0.578		0.554	0.458
f_{observed}	0.239	0.522		0.511	0.554

C insertions in hotspots

$p < 10^{-150}$	-2 C	-1 R	CCC	+1 A	+2 Y
N	463	955	1047	1045	1012
f_{expected}	0.198	0.624		0.473	0.527
f_{observed}	0.442	0.912		0.998	0.967
$p = 7.0 \times 10^{-9}$	-2 C	-1 R	CCCC	+1 A	+2 Y
N	97	317	453	310	280
f_{expected}	0.194	0.617		0.506	0.531
f_{observed}	0.214	0.700		0.684	0.618
$p = 3.9 \times 10^{-5}$	-2 C	-1 R	CCCCC	+1 A	+2 Y
N	81	165	259	164	165
f_{expected}	0.231	0.637		0.509	0.514
f_{observed}	0.313	0.637		0.633	0.637

all A insertions

$p = 6.1 \times 10^{-81}$	-2 T	-1 C	AAAA	+1 Y	
N		412	738	1172	1028
f_{expected}		0.299	0.338		0.684
f_{observed}		0.352	0.630		0.877
$p = 3.4 \times 10^{-16}$	-2 T	-1 C	AAAAA	+1 Y	
N		143	224	401	241
f_{expected}		0.302	0.329		0.676
f_{observed}		0.357	0.559		0.601
$p = 1.8 \times 10^{-4}$	-2 T	-1 C	AAAAAA	+1 Y	
N		98	148	395	321
f_{expected}		0.303	0.327		0.679
f_{observed}		0.248	0.375		0.813

A insertions in hotspots

$p = 1.3 \times 10^{-23}$	-2 T	-1 C	AAAA	+1 Y	
N		57	99	120	109
f_{expected}		0.299	0.338		0.684
f_{observed}		0.475	0.825		0.908

$p = 1.5 \times 10^{-4}$	-2 T	-1 C	AAAAA	+1 Y
N	18	27	40	25
f_{expected}	0.302	0.329		0.676
f_{observed}	0.450	0.675		0.625

The legend for Supplemental Table 5 has now been expanded to clearly explain the analysis and its interpretation. The legend now reads: “The significance of motifs flanking homopolymers with insertions in the *cdc9-EEAA msh2Δ* strain. Base identities and positions (e.g. ‘-2 C’) are all given relative to the homopolymer oriented 5'-to-3' in the lagging template strand. N denotes the number of insertions observed with the base identity. The count under the homopolymer (e.g. ‘CCC’) is the total number of insertions observed in homopolymers of that type and length (N_{total}). Expected fractions (f_{expected}) were calculated from the flanks of all homopolymers of the given type and length in the 26% of the genome where the leading/lagging strand template is known for >90% of replications from ribonucleotide mapping [Zhou, 2019]. The p -values are from the Chi-squared tests of observed stated versus expected counts ($N_{\text{total}} \times f_{\text{expected}}$).

- The first part of my original question “Why is the repeat length peak for insertions different for T/A insertion than for G/C insertions? Is there a structural explanation for this (considering the structure presented in the 2021 paper which suggests that the bulge is accommodated in a specific alignment register)? Also, do the authors have a (structural) explanation or hypothesis for why there is a preference for G>C>T>A?” was not responded to, and I understand and except that the second part will be addressed in future work. This is not essential, but perhaps these could be briefly commented on (e.g. in the discussion) as outstanding questions?

Response: We apologize for not satisfactorily addressing the first part of your question in our revised manuscript. We agree that these issues are important to comment on and now include the following in the Discussion on pages 13-14: “ Furthermore, the reason for repeat lengths being different for T•A insertions than for C•G insertions (Fig. 4-5) is intriguing and warrants further investigation. The data indicate an order of G>T>C>A and both repeat tract length and the identity of the inserted nucleotide could be dictated by the fidelity of the DNA polymerase during synthesis, the fidelity of DNA ligase during nick sealing, or some combination of both. These outstanding questions are of great interest and will require detailed biochemical analysis.